# Antibodies utilizing VL6-57 light chains target a convergent cryptic epitope on SARS-CoV-2 spike protein and potentially drive the genesis of Omicron variants

Qihong Yan [1,9], Xijie Gao [2,9], Banghui Liu[3,9], Ruitian Hou[4], Ping He[5], Yong Ma[3], Yudi Zhang[3,6], Yanjun Zhang[1], Zimu Li[3], Qiuluan Chen[7], Jingjing Wang[3], Xiaohan Huang[1], Huan Liang[1], Huiran Zheng[1], Yichen Yao[8], Xianying Chen[1], Xuefeng Niu[1], Jun He [3] ✉, Ling Chen [1,3,5] ✉, Jincun Zhao [1,5,8] ✉ & Xiaoli Xiong [3] ✉

Continued evolution of SARS-CoV-2 generates variants to challenge antibody immunity established by infection and vaccination. A connection between population immunity and genesis of virus variants has long been suggested but its molecular basis remains poorly understood. Here, we identify a class of SARS-CoV-2 neutralizing public antibodies defined by their shared usage of VL6-57 light chains. Although heavy chains of diverse genotypes are utilized, convergent HCDR3 rearrangements have been observed among these public antibodies to cooperate with germline VL6-57 LCDRs to target a convergent epitope defined by RBD residues S371-S373-S375. Antibody repertoire analysis identifies that this class of VL6-57 antibodies is present in SARS-CoV-2-naive individuals and is clonally expanded in most COVID-19 patients. We confirm that Omicron-specific substitutions at S371, S373 and S375 mediate escape of antibodies of the VL6-57 class. These findings support that this class of public antibodies constitutes a potential immune pressure promoting the introduction of S371L/F-S373P-S375F in Omicron variants. The results provide further molecular evidence to support that antigenic evolution of SARS-CoV-2 is driven by antibody mediated population immunity.

The SARS-CoV-2 pandemic offered an opportunity to interrogate immune response towards an emergent virus infecting an immunologically naive population. During the first 3 years of the pandemic, the spike (S) protein on the virus surface has been actively undergoing antigenic drift generating a large number of variants[1–4]. In late 2021, the Omicron BA.1 variant emerged, its highly mutated S-protein renders most antibodies isolated earlier in the pandemic ineffective, conferring the Omicron BA.1 variant with an extraordinary immune evasion capability[5,6]. Further antigenic drift of S-protein had been

observed and numerous Omicron subvariants emerged, including BA.2-BA.5[7–9]. Since late 2022, XBB and BQ.1 subvariant series have emerged based on BA.2 and BA.5 respectively with further antigenic changes on the S-protein[10,11]. As the pandemic progresses, S-protein is drifting antigenically further generating new variants.

The emergence of Omicron variants marked great shifts in S-protein antigenicity and virological behaviors, sparking several theories on origins of Omicron variants[12]. Several recurrent mutations observed in earlier variants became fixed in the Omicron S-protein,

most of these are located within the receptor binding domain (RBD) of the S-protein, including K417N, L452R, T478K, E484A and N501Y. Earlier studies have identified that some of these mutations are located in the epitopes recognized by several classes of germline antibodies that are widely induced within the population. K417N is located in the epitope of VH3-53/3-66 encoded class 1 antibodies inhibiting ACE2 binding[13–15]; L452R is located in the epitope of VH1-69 encoded population antibodies[16–18]; E484A is located in the epitope of VH1-2 encoded class 2 population antibodies[19–21]. These mutations often confer viruses with considerable antibody evading ability. While L452R and N501Y have also been proposed to enhance or compensate fusogenic or receptor binding activity of S-protein[4,22–24], association of recurrent mutations to population antibody responses strongly implies that antigenic drift of SARS-CoV-2 S-protein may be driven by immune pressure at population level. Population immunity (or herd immunity) has long been proposed to drive antigenic drift of influenza virus haemagglutinin[25], despite molecular basis behind such driving force remains poorly characterized. Apart from the above identified recurrent mutations, several RBD mutations were first emerged with the highly mutated Omicron BA.1 S-protein, including S371L, S373P, S375F, N440K, G446S, S477N, Q493R, G496S and Q498R, most of them have been fixed in the S-protein of current circulating viruses. Among these, S371L (S371F in BA.2-BA.5), S373P and S375F are particularly interesting that they are close together in sequence while located relatively far away from the receptor binding site[26]. The S371L/F-S373P-S375F mutations are considered as an important feature of Omicron variants and have been recently shown as an important determinant for the Omicron nasal cell tropism[27]. They have also been shown to modulate S-protein functions[28,29] and antigen presentation[30]. It has been hypothesized that the altered Omicron tissue tropism can also affect disease severity[31–34]. However, the driving force behind the genesis of the S371L/F-S373P-S375F mutations remains enigmatic. In this study, we report that diverse antibodies utilizing IGLV6-57 (VL6-57) light chains are widely induced within the population and they target a common epitope defined by S371, S373 and S375 in the ancestral SARS-CoV-2 S-protein. We provide evidence to suggest that the identified class of antibodies constitute a potential immune pressure driving the genesis of S371L/F-S373P-S375F mutations in Omicron variants.

## Results

### Identification of SARS-CoV-2 S-specific antibodies with VL6-57 encoded light chains

In an exercise to isolate SARS-CoV-2 S-specific antibodies from convalescent patients infected at the beginning of the pandemic (early 2020), we have reported the isolation of 6 mAbs, namely R1-26, R1-30, R1-32, R2-3, R2-6, and R2-7 by phage display using SARS-CoV-2 RBD as the bait[16]. Interestingly, five out of the six isolated antibodies utilize light chains encoded by IGLV6-57, while their heavy chains are of different genetic origins (Fig. 1a). The identified antibodies either have zero or very low somatic hypermutation rates (Fig. 1a), suggesting that they are germline antibodies. Notably, 4 mAbs, namely R1-30, R2-3, R2-6, and R2-7 have a shared "WLRG" motif in the middle of their HCDR3 (Fig. 1a). Different from the other 4 isolated mAbs, R1-26 has a hydrophobic "LGPWV" motif in the middle of HCDR3 (Fig. 1a). The five isolated antibodies with IGLV6-57 light chains have affinities in the range of 3.8-62.2 nM towards the wildtype SARS-CoV-2 RBD (Fig. 1a), with R1-26 being the strongest ($K_D$ = 3.8 nM). In binding competition assays, we found that all the 5 mAbs compete with ACE2 and with each other to bind RBD (Fig. 1b). These results suggest that all the 5 mAbs target overlapping epitopes. In pseudovirus neutralization assays, their neutralization activities largely correlate with their affinities, the strongest binder R1-26 has an $IC_{50}$ as low as 2.7 nM (Fig. 1a). We studied the most potent mAb-R1-26 in detail, and we found that R1-26 is able to bind RBD of SARS-CoV-2 variants emerged before the Omicron BA.1 variant (Fig. 1c, Supplementary Table 1). Consistently, R1-26 is able to

bind the corresponding S-trimers tightly without dissociating, likely through avidity (Fig. 1d, Supplementary Table 1). We further confirmed that R1-26 has neutralization activity towards wildtype, Alpha, Beta, and Delta SARS-CoV-2 authentic viruses in cell culture (Fig. 1e).

### Binding of R1-26 antibody to SARS-CoV-2 S-trimer

To further understand neutralization activity of R1-26, we determined cryo-electron microscopy (cryo-EM) structures of S-trimer:R1-26 Fab complexes in two different stoichiometries (3:2, 3:3 S-protomer:R1-26 Fab) (Figs. 2a, S1a) with upper resolutions in the range of 3.19-3.44 Å (Fig. S2a–c). An unusual structure showing a head-to-head aggregate of two S-trimers each bound by 3 R1-26 Fabs was also determined at a lower resolution of 5.3 Å (Figs. 2a, S1a, S2a–c), the aggregation is mediated by interactions between Fabs bound to different S-trimers and a similar aggregate was observed for the S-trimer:IgG complex of 6M6 (using IGHV3-9, IGLV3-21 genes)[35]. It is not known whether the observed interactions could lead to aggregation of S-trimers on virus surface.

Based on a locally refined RBD:Fab structure at ~3.5 Å resolution derived from the 3:3 S-protomer:R1-26 Fab complex dataset (Figs. S1a, S2a), the detailed R1-26 epitope is resolved (Fig. 2b). Buried surface area (BSA) analysis reveals that HCDRs and LCDRs of R1-26 bury comparable RBD surface areas–459.0 Å² and 385.7 Å² respectively. Among CDR loops, HCDR3 dominates the contact with RBD, the hydrophobic HCDR3 loop, containing residues L101, G102, P103 and W104, probes into a hydrophobic cavity formed by RBD residues Y369, A372, F374, F377 and P384 (Figs. 2b left panel, S2d). This hydrophobic contact is further stabilized by surrounding electrostatic and polar interactions, distributed in two different patches: the larger interaction patch is mediated by hydrogen bonds between backbone of RBD residues S375, F377 and LCDR1 residue N32 sidechain; a cation-π interaction between RBD residue K378 and LCDR1 residue Y33; a charged hydrogen bond between RBD residue T385 and LCDR2 residue E51; and finally a salt bridge between RBD residue K378 and LCDR2 residue D52 (Fig. 2b, right panel). The second interaction patch is mediated by a hydrogen bond between the glycan attached to RBD residue N343 and HCDR2 residue K52; and a hydrogen bond between RBD residue N370 and HCDR2 residue Q53 (Fig. 2b, right panel). Finally, RBD residue A372 is contacted by LCDR3 residue Y94 (Fig. 2b, right panel).

The detailed R1-26 epitope structure confirms that the epitope is fully buried in the previously determined RBD "down" closed and locked S-trimer structures[36–38], therefore, the epitope is fully cryptic in a S-trimer adopting 3-RBD "down" closed or locked conformations (Fig. S3a–h). In the 1 RBD "up" S-trimer, this epitope on the "up" RBD is also largely obstructed (Fig. S3a). Likely due to incompatibility with 3-RBD "down" and 1 RBD "up" S-trimer conformations, at least 2 RBDs are observed to adopt "up" position when R1-26 is bound to a S-trimer and only 3:2, 3:3 S-protomer:Fab structures are observed when R1-26 Fab was added in excess (Fig. 2a). The detailed epitope structure also confirmed that the R1-26 epitope does not overlap with the ACE2 binding interface (Fig. S3f, h), indicating that R1-26 Fab is a class 4 antibody (Fig. S3f–h). CR3022 is a well-studied class 4 mAb which cross-reacts to RBDs of SARS-CoV-1 and SARS-CoV-2[39]. We found that although the epitope of R1-26 appears to largely overlap with that of CR3022, R1-26 features a different approach angle towards the RBD compared with CR3022 (Fig. S3a, g, h). Notably, the canonical class 4 antibody CR3022 does not inhibit ACE2 binding and is only weakly neutralizing[16,40]. In contrast, strong ACE2 binding inhibition is observed for R1-26 (Fig. 1b) and other class 4 antibodies such as C118 and C022[40], as well as S2A4 and S2X35[41]. By modeling, we found that the approach angle of R1-26 is more tilted towards the modeled ACE2 bound to RBD (Fig. S4a), while the approach angle of CR3022 is more tilted away from it (Fig. S4b). In addition, R1-26 and CR3022 adopt different orientations when bound to RBD (Fig. S4a, b). Due to the

**a Features of isolated IGLV6-57 (VL6-57) light chain utilizing mAbs**

| Donor | mAb | Gene analysis | | | | | | | | | Binding of mAb to RBD | Pseudovirus neutralization |
| | | IGHV | IGHD | IGHJ | HCDR3 | SHM% | IGLV | IGLJ | LCDR3 | SHM% | $K_D$ (nM) | $IC_{50}$ (nM) |
|---|---|---|---|---|---|---|---|---|---|---|---|---|
| PtK | R1-26 | IGHV3-7 | IGHD6-6 | IGHJ4 | ARGQLGPWVGVDY | 0 | IGLV6-57 | IGLJ3 | QSYDSSNWV | 0 | 3.84 | 2.71 |
| PtK | R1-30 | IGHV4-59 | IGHD4-23 | IGHJ4 | ARQGWLRGNFDY | 0 | IGLV6-57 | IGLJ2 | QSYDSSIHVV | 0 | 35.38 | 4.06 |
| PtZ | R2-3 | IGHV3-7 | IGHD5-18 | IGHJ3 | ASQLWLRGAFDI | 2.78 | IGLV6-57 | IGLJ3 | QSYDSSNPWV | 1.72 | 13.28 | 3.46 |
| PtZ | R2-6 | IGHV4-31 | IGHD5-24 | IGHJ3 | ARKGWLRGAFDI | 1.37 | IGLV6-57 | IGLJ2 | QSYDSGVV | 0.69 | 22.8 | 13.91 |
| PtZ | R2-7 | IGHV4-31 | IGHD5-24 | IGHJ3 | ARKGWLRGAFDI | 1.37 | IGLV6-57 | IGLJ2 | QSYDSSNHLVV | 0 | 62.2 | >333 |

**b RBD binding competition assays**

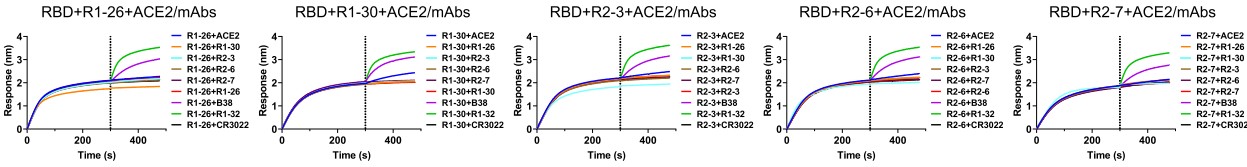

**c RBD binding by R1-26**

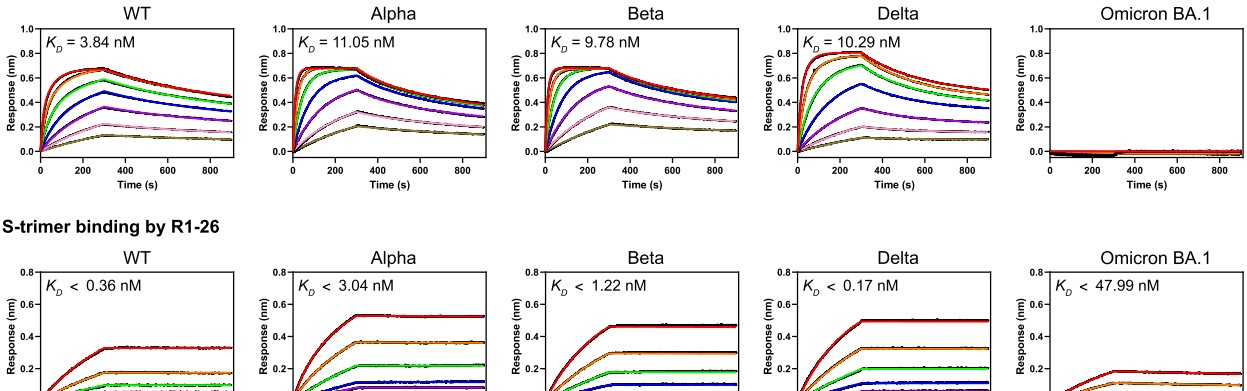

**d S-trimer binding by R1-26**

**e Authentic virus neutralization by R1-26**

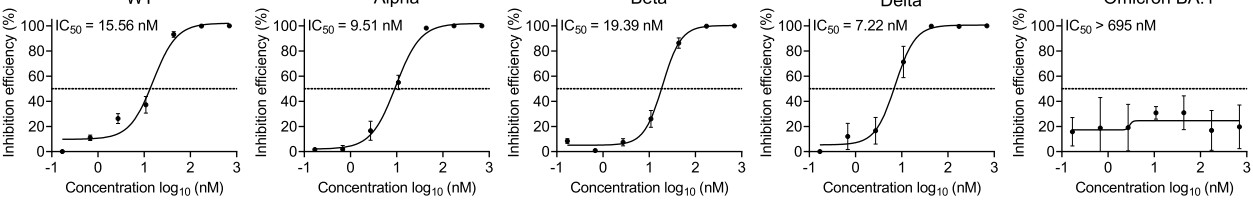

**Fig. 1 | Characteristics of the 5 VL6-57 light-chain utilizing antibodies. a** The genetic and functional properties of the five isolated VL6-57 light chain utilizing antibodies, as previously reported[16]. Neutralization titers ($IC_{50}$) of antibodies were determined against a pseudovirus displaying WT SARS-CoV-2 S-protein. Binding affinities ($K_D$) of antibodies to SARS-CoV-2 WT RBD were determined by biolayer interferometry (BLI) assays. Germline usage and somatic hypermutation (SHM) analysis of antibodies were performed using IMGT/V-QUEST. **b** Pairwise binding competition to SARS-CoV-2 RBD was assessed by BLI. A biosensor immobilized with WT SARS-CoV-2 RBD was first saturated with one of the VL6-57 utilizing mAbs until the dashed line before submerging into a solution of another mAb or ACE2 to assess competition. An ACE2 competing antibody B38 (purple) and two non-ACE2 competing antibodies recognizing different RBD epitopes, CR3022 (black)[39] and R1-32 (green)[16] were used as controls. **c, d** BLI binding curves of R1-26 to 2-fold serially diluted RBD (**c**) or S-protein (**d**) solutions of WT SARS-CoV-2 and VOCs. The black lines represent the experimentally recorded sensorgram traces, the colored lines represent corresponding fits. Detailed binding kinetics parameters are summarized in Supplementary Table 1. **e** Neutralization activities of R1-26 towards SARS-CoV-2 WT and VOC authentic viruses in cell culture (data are presented as mean values ± SD, $n = 3$). Source data for **e** are provided as a Source Data file.

differences in RBD binding, the simultaneous binding of R1-26 and ACE2 to the RBD would result in steric clashes between R1-26 and the glycan chain attached to ACE2 residue N322 (Fig. S4a). By contrast, CR3022 and ACE2 are able to bind the RBD simultaneously without clashing (Fig. S4b). This is consistent with a previous report indicating that CR3022 does not block ACE2 binding[42]. Differences in RBD binding between R1-26 and CR3022 likely confer superior ACE2 blocking activity to R1-26. A few studies have also linked the neutralization activities of class 4 antibodies (including VL6-57 antibodies S2A4) to their ability to block ACE2 binding[40,41].

We further found that fusogenic competent native S-R trimer (without proline stabilization) incubated with R1-26 undergoes structural transition into a post-fusion conformation (Fig. 2c). This analysis shows that R1-26 possesses an activity to trigger fusogenic conformational change similar to class 1 mAbs including B38[16] (Fig. 2c) and S230[43]. Interestingly, the class 4 antibody CR3022 does not have this triggering activity (Fig. 2c). Both ACE2 binding inhibition and premature triggering of spike fusogenic change by R1-26 could lead to the inhibition effect observed in a S-protein-ACE2 interaction mediated cell-cell membrane fusion assay by which cell entry inhibition activity

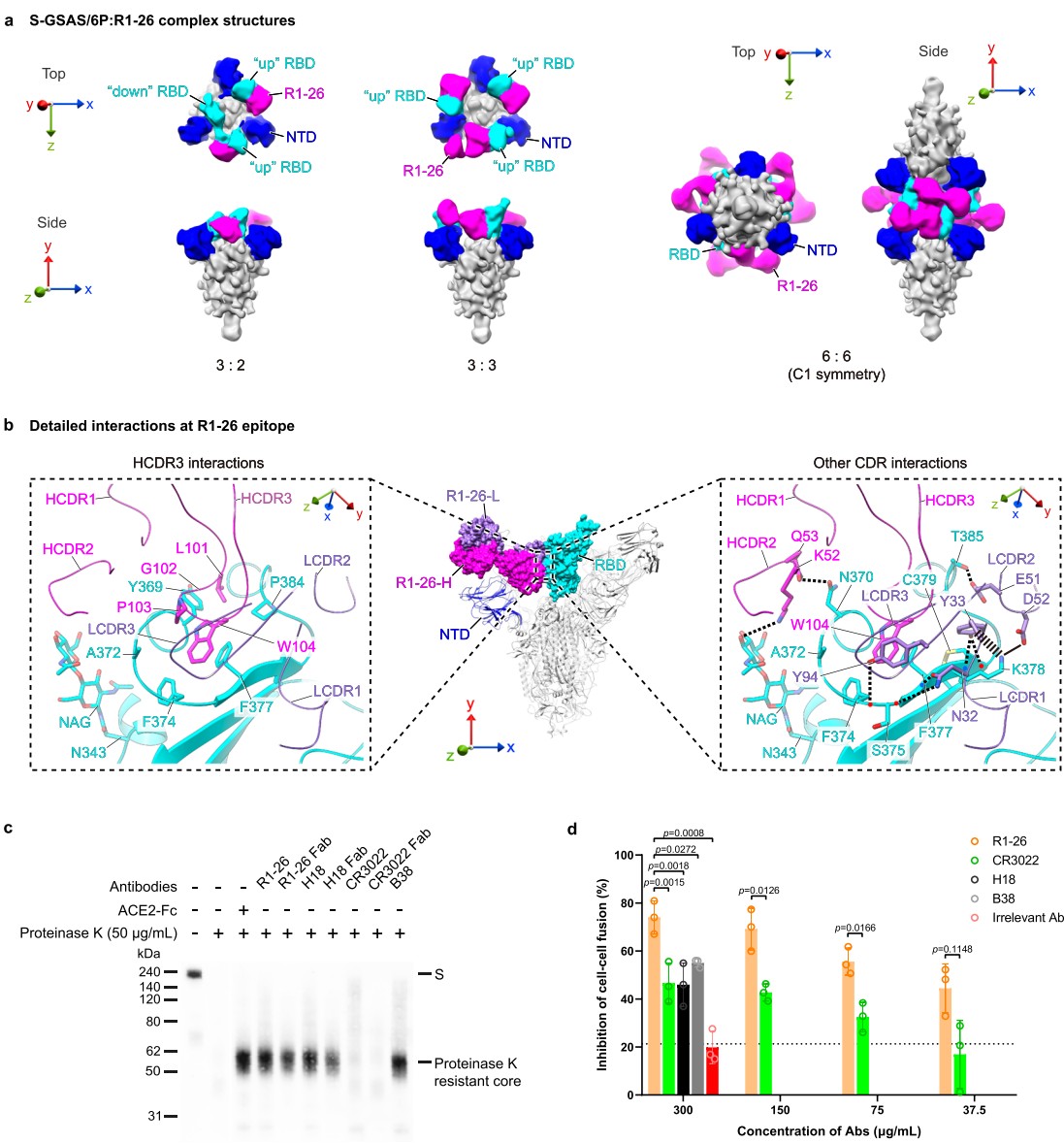

**Fig. 2 | Structural analysis of R1-26 epitope binding and biochemical activities of R1-26. a** Structures (low-pass filtered to 12 Å) of S-GSAS/6P S-trimers in complex with R1-26 Fabs in different stoichiometries. R1-26 Fab, NTD and RBD are highlighted in magenta, blue and cyan, respectively; the rest of the S-trimer is colored gray. **b** Epitope of R1-26 on spike RBD. R1-26-H and R1-26-L chains are colored in magenta and purple. Detailed interactions between R1-26 and RBD are shown in the dashed boxes. CDR loops are indicated, selected interacting residues in antibody-RBD interface are shown. Thick and thin dashed lines indicate cation-π and hydrogen bond interactions; The solid line represents a salt bridge. **c** Ligand-induced conformational change assays to probe the induction of post-fusion structures, presence of proteinase K resistance core is indicative of post-fusion structure[16,43,103]. **d** R1-26 inhibits spike-ACE2 interaction mediated cell-cell membrane fusion (data are presented as mean values ± SD, n = 3). Two-tailed paired student's t test was performed. Source data for **c** and **d** are provided as a Source Data file.

of antibody can be assessed[44] (Fig. 2d). Both activities likely contribute to the neutralization activity of R1-26.

## VL6-57 light chain can pair with diverse heavy chains to bind a convergent epitope

To further understand antigen binding by VL6-57 light-chain utilizing mAbs, we surveyed the Protein Data Bank (PDB) for structurally characterized VL6-57 light-chain utilizing mAbs bound to SARS-CoV-2 S-protein or RBD. Among a dataset of 376 SARS-CoV-2 S-specific mAb complex structures, which are all experimentally determined by X-ray crystallography or cryo-EM, 12 mAbs utilize VL6-57 encoded light chains, of note, among them, 10 belong to class 4 antibodies (see Fig. S5 for a structure gallery of class 4 antibodies), these account for more than a fourth of class 4 antibodies (n = 35) found amongst the

376 structurally characterized SARS-CoV-2 S-specific mAbs (Fig. S5). Further, germline usage analysis shows that VL6-57 is the most frequently used light-chain germline gene among the class 4 antibodies (Fig. S6a, left panel). An enrichment of VL6-57 is also observed for the F2 group antibodies (equivalent to class 4 antibodies, as defined by deep mutational scanning)[7] (Fig. S6a, right panel). We did not find evidence of preferential VL6-57 gene usage in the global B cell repertoire data of SARS-CoV-2-naive individuals[45]. The above analyses strongly suggest that VL6-57 gene preferentially generates S-specific class 4 mAbs, leading to the hypothesis that VL6-57 light chains may play an important role in antigen binding.

We analyzed buried surface area (BSA) to further understand the contribution of VL6-57 light chains in epitope binding. Among the 376 structurally characterized SARS-CoV-2 S-specific mAbs, heavy

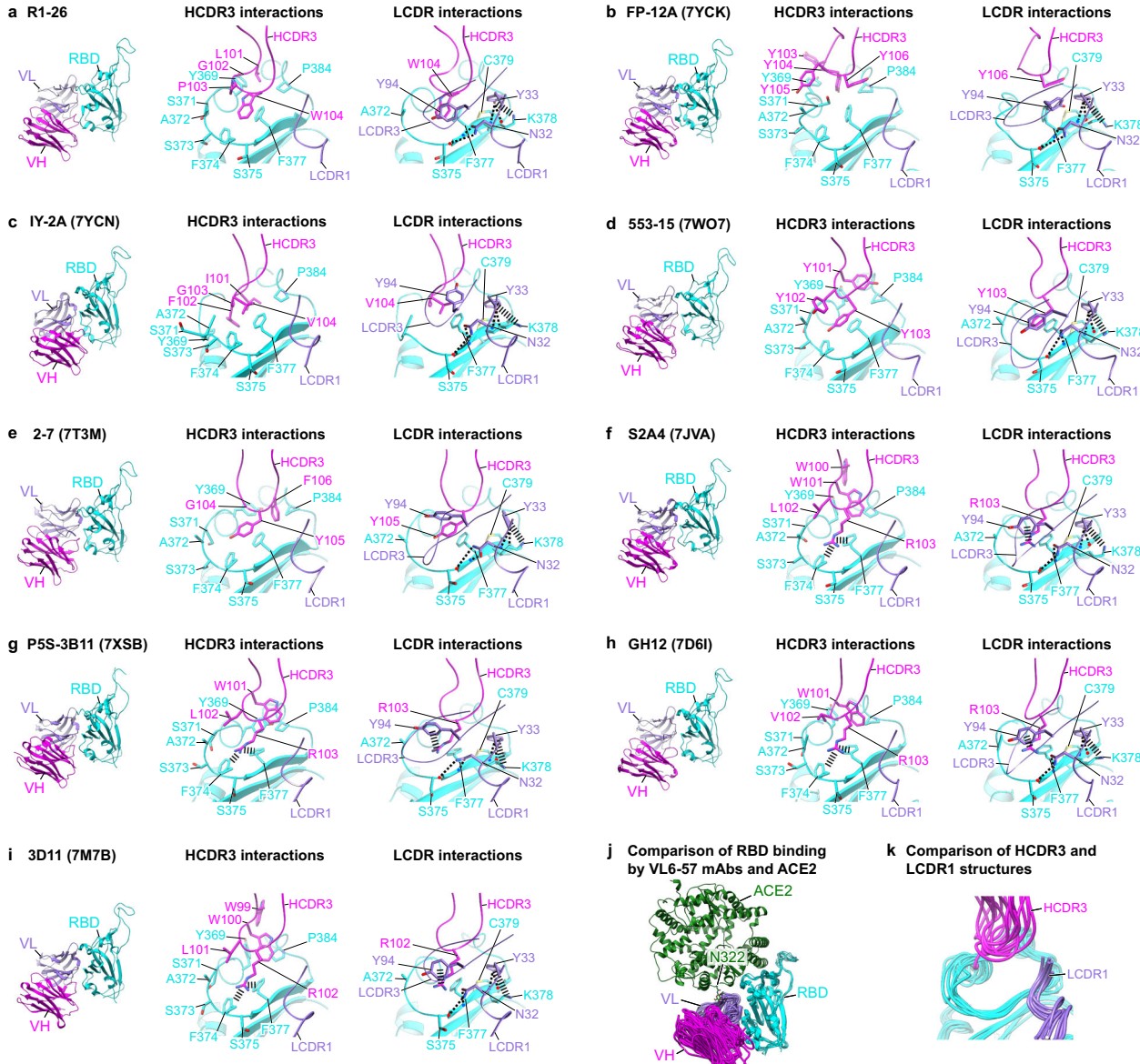

**Fig. 3 | Epitopes and binding modes of VL6-57 antibodies. a–i** Structures of R1-26 (**a**), FP-12A (**b**), IY-2A (**c**), 553-15 (**d**), 2-7 (**e**), S2A4 (**f**), P5S-3B11 (**g**), GH12 (**h**), and 3D11 (**i**) bound to their epitopes. Fab-VH, Fab-VL, and RBD are colored in magenta, purple, and cyan, respectively. Selected interactions engaged by HCDR3, LCDR1, and LCDR3 are shown. All shown antibodies bind almost the same surface area on RBD, hydrophobic, particularly aromatic amino-acid residues are found in all the HCDR3 loops of the shown VL6-57 antibodies for epitope recognition. All LCDR1 loops have the same interactions with RBD. **j** Comparison of RBD binding by the shown VL6-57 mAbs and ACE2. **k** Superposition of HCDR3 and LCDR1 loops of the shown VL6-57 mAbs. Variability is observed for the HCDR3 loops while the LCDR1 loops show high similarity.

chains dominate epitope interaction, burying significantly more surface areas than light chains (HC vs. LC: $613 \pm 173$ Å$^2$ vs. $283 \pm 149$ Å$^2$) (Fig. S7a). Different from typical SARS-CoV-2 S-specific mAbs, the average BSA of VL6-57 class 4 mAbs is comparable between heavy and light chains (HC vs. LC: $466 \pm 165$ Å$^2$ vs. $418 \pm 104$ Å$^2$) (Fig. S7b). This analysis suggests that among VL6-57 class 4 mAbs, light chains are at least equally important as heavy chains in epitope recognition. Further analysis reveals that LCDR1 and HCDR3 contribute more than 60% of BSA (Fig. S7c).

Superposition of the 10 VL6-57 class 4 Fab:RBD complex structures reveals that apart from mAbs–10-40 and 002-13 (Fig. S5), all the other 8 mAbs bind to epitopes that are almost identical to the epitope of R1-26 (Figs. 3a–k, S5), demonstrating clear epitope convergence among VL6-57 mAbs. Comparison of light and heavy-chain CDR sequences reveals that all these structurally characterized mAbs share similar LCDR1-3 sequences (Fig. S6c). Interestingly, structurally

characterized S2A4, P5S-3B11, GH12, and 3D11 not only exhibit high similarity in LCDR sequences (Fig. S6c) to the isolated R1-30, R2-3, R2-6, and R2-7 mAbs, they also feature 12-residue long HCDR3s with either "WLRG" or "WVRG" motif in the middle (Fig. S6b). Therefore, R1-30, R2-3, R2-6, and R2-7 are likely to bind the same epitope as S2A4, P5S-3B11, GH12, and 3D11, which is almost identical to that of R1-26, consistent with the result that R1-26 competes with R1-30, R2-3, R2-6, and R2-7 in binding to spike RBD (Fig. 1b). The sequence analysis also identifies that VL6-57 light chains can also pair with heavy chains with HCDR3 loops rich in hydrophobic, particularly aromatic amino acids. mAbs 553-15, 2-7, FP-12A and IY-2A feature "WYYY", "GYFY", "YYYY", and "LGIFG" motifs in the middle of HCDR3 respectively, notably, the "LGIFG" in IY-2A is reminiscent of the "LGPWV" motif found in R1-26 HCDR3. Analysis of their structures shows that these mAbs bind to the same epitope as for S2A4, P5S-3B11, GH12, and 3D11 with "WLRG" or "WVRG" HCDR3 motifs (Figs. 3, S5, S6b).

Detailed structural comparison of the identified VL6-57 mAb:RBD complexes reveals that HCDR3s in R1-26, S2A4, 3D11, GH12, P5S-3B11, 553-15, 2-7, FP-12A and IY-2A interact with almost identical RBD residues within the convergent epitope with several modes of interactions (Fig. 3a–j). R1-26, FP-12A, and IY-2A (Fig. 3a–c) appear to bind the epitope using hydrophobic contacts primarily from aromatic residues. 553-15 and 2-7 (Fig. 3d, e) bind the epitope by π-π interactions (from the HCDR3 tyrosine) combined with hydrophobic contacts. Finally, S2A4, P5S-3B11, GH12 and 3D11 (Fig. 3f–i) binds the epitope with a combination of cation-π interactions (by the arginine in HCDR3) and hydrophobic contacts (by the tryptophan in HCDR3). Different binding modes give rise to variability among HCDR3 structures (Fig. 3k), however, hydrophobic interactions are shared among these different interaction modes reflecting the convergent epitope being hydrophobic, containing multiple aromatic residues including Y369, F374 and F377 (Fig. 3a–i).

The comparison also identifies that the germline-encoded N32 and Y33 from LCDR1 engage in conserved interaction to spike RBD among the 8 VL6-57 light-chain utilizing R1-26-like mAbs (Figs. 3a–i, S6c), and LCDR1s show little variability when binding to RBD (Fig. 3k). BSA analysis also confirms that LCDR1s bury the largest surface areas among LCDRs (Fig. S7c), suggesting that this germline-encoded "NY" motif facilitates efficient RBD binding. For LCDR2, the germline-encoded E51 and D52 shared among the 8 R1-26-like mAbs form conserved interactions to RBD residues T385/S383 and K378 respectively, as observed in R1-26 (Figs. 2b, 5f, S6c, S7c). Collectively, our results demonstrate that the VL6-57 light chain likely serves as an efficient framework for the generation of class 4 SARS-CoV-2 S-specific mAbs.

## VL6-57 antibodies paired with HCDR3 "WLRG" motif are widely induced in the population and clonally expanded upon SARS-CoV-2 infection

The presence of considerable amount of VL6-57 antibodies among the structurally characterized class 4 mAbs prompted us to further search CoV-AbDab (the Coronavirus Antibody Database)[46] for additional VL6-57 mAbs to investigate their features (Fig. 4a, see Method). The search obtained 290 mAbs utilizing VL6-57 light chains from 49 independent studies. A total of 36 different heavy-chain germline genes are observed to pair with VL6-57 to generate S-specific mAbs (Fig. S8a). Notably, we found that there is an enrichment of mAbs with 12-AA long HCDR3s and 9-10-AA long LCDR3s, accounting for 34% (99/290) of the curated VL6-57 mAbs (Fig. S8b–d). Moreover, those VL6-57 mAbs with 12-AA long HCDR3s appear to be preferentially derived from heavy-chain genes IGHV3-7, IGHV4-39, and IGHV4-59 (Fig. 4b). HCDR3 sequence analysis shows that there is a strong preference for a 12-AA long HCDR3 containing the "WLRG" motif (68/99 mAbs) as observed in S2A4, P5S-3B11, 3D11, R1-30, R2-3, R2-6 and R2-7 (Figs. 4b, S6b). A previous survey also noticed that the isolated SARS-CoV-2 S-specific VL6-57 antibodies tend to have a "WLRG" motif at the tip of the HCDR3[47]. In addition, we also found a small percentage of mAbs (7/99) containing a 12-AA long HCDR3 with the "YYY" motif observed in 553–15 (Fig. 4b logo plot, S6b). However, we failed to identify consensus HCDR3 sequences among VL6-57 mAbs with 11-, 13-, 14-, or 15-AA long HCDR3s (Fig. S8e). V(D)J rearrangement analysis suggests that the W, L, and R residues within the HCDR3 "WLRG" motif are most likely encoded by IGHD5-12 gene when its 2nd reading frame is used (Fig. S8f). The last AA residue G within the motif is located at the DJ junction and is most likely a result of an insertion at the C-terminal side of the D segment (Fig. S8f). These findings explain the genetic origin of the HCDR3 "WLRG" motif. An LCDR3 sequence analysis reveals that a "QSYDSS" motif is enriched (Fig. S8d). A BSA analysis indicates that LCDR3s cover smaller areas by comparison with LCDR1s in VL6-57 mAbs, suggesting that LCDR3s may play an auxiliary role in antigen binding (Fig. S7c). By structural analysis, we found that the tyrosine (Y94 in R1-26, Fig. 2b) residue within the "QSYDSS" motif is engaging in specific interactions

with critical antigen binding residues in HCDR3. In R1-26, FP-12A, IY-2A, 553-15 and 2-7 the LCDR3 tyrosine is interacting with antigen contacting HCDR3 residue W104, Y106, V104, Y103 or Y105, respectively, by hydrophobic contacts (Fig. 3a–e, LCDR interaction panels). In S2A4, P5S-3B11, GH12, and 3D11, the LCDR3 tyrosine is interacting with the R in the HCDR3 "WLRG" motif by cation-π interactions (Fig. 3f–i, LCDR interaction panels). These specific interactions suggest LCDR3 supports HCDR3 in antigen binding.

The above analysis suggests that the "WLRG" motif within the 12-AA long HCDR3 and the "QSYDSS" motif within the LCDR3 are the convergent signatures of the SARS-CoV-2 RBD-specific VL6-57 mAbs, which may be widely induced in the COVID-19 population. Such proposal may explain previous observations that antibodies utilizing the light-chain gene IGLV6-57 are over-represented in the isolated SARS-CoV-2 S-specific antibodies[45,47,48]. To verify this, we next searched our previously published bulk antibody repertoires from 24 SARS-CoV-2 naive donors (ND) and 33 COVID-19 convalescents[13,49,50] (Fig. 4a, see Methods). The COVID-19 donors PtZ and PtK, from whom R1-26, R1-30, R2-3, R2-6 and R2–7 have been isolated, are also included in this analysis. In addition to IgH and IgL sequences identical to the 5 isolated mAbs, similar "WLRG" motif containing IgH sequences and "QSYDSS" motif containing VL6-57 IgL sequences can be readily detected in the repertoires of PtZ and PtK (Fig. S9). The search shows that "WLRG" motif containing IgH sequences can be detected in 29/33 COVID-19 convalescents and 10/24 naive donors (Fig. 4c). Following SARS-CoV-2 exposure, there was a remarkable enrichment of "WLRG" motif containing IgH sequences (Fig. 4d). We observed clonal expansion of the "WLRG" motif containing IgH sequences in COVID-19 patients by tracking the longitudinal samples collected from 4–28 days post symptom onset (Fig. 4e, f). It is worth mentioning that the "WLRG" motif containing IgH sequences present in the SARS-CoV-2-exposed antibody repertoires are mainly IgG isotype (encoded by IGHG, 79.3%). However, 65.2% of the "WLRG" motif containing IgH sequences detected in the SARS-CoV-2-naive antibody repertoires are IgM isotype (encoded by IGHM) that is usually expressed by naive B cells (Fig. 4g, h). Unlike heavy chains, the "QSYDSS" motif containing VL6-57 light chains can be readily detected in both the naive and SARS-CoV-2-exposed IgL repertoires, highlighting abundance of VL6-57 transcripts in human B cell repertoires even under resting state (Fig. 4i).

To further determine the abundance of B cells expressing mAbs of VL6-57 light chains paired with the HCDR3 "WLRG" motif, we analyzed published single-B V(D)J sequences from SARS-CoV-2-exposed and -naive individuals[51–55] and our previously reported FACS-sorted RBD-reactive single-B memory cell sequences from COVID-19 convalescents[56] (Fig. 4a, see Method). Interestingly, only two VL6-57 B cells with paired "WLRG" motif (2 IGHM) were detected among 87808 B cells (0.02‰) from SARS-CoV-2-naive donors (Fig. 4j). Following SARS-CoV-2 vaccination or infection, there is a remarkable expansion of B cells expressing VL6-57 antibodies with paired "WLRG" motif. We identified 20 VL6-57 B cells with paired "WLRG" motif (15 IGHG, 4 IGHM, and 1 IGHA) among 396221 B cells (0.05‰) from COVID-19 patients, while 14 (12 IGHG, 1 IGHM, and 1 IGHA) were identified among 29838 B cells (0.47‰) from COVID-19 vaccinees. Furthermore, 14 (13 IGHG and 1 IGHM) were found among the 4642 FACS-sorted RBD-reactive B cells (3.0‰) from COVID-19 convalescents (Fig. 4j). After FACS sorting, there is an approximately 60-fold (3‰/0.05‰) enrichment of VL6-57 B cells with paired "WLRG" motif among the RBD-reactive B cells compared with the unsorted single-B cell sequencing data (Fig. 4j). Consistently, the VL6-57 B cells with paired "WLRG" motif identified from SARS-CoV-2-naive individuals are mostly expressing IgM, while those from COVID-19 patients and vaccinees are predominantly expressing IgG (Fig. 4j). In summary, the above observations indicate that naive B cells expressing VL6-57 antibodies with paired "WLRG" motif were activated after SARS-CoV-2 exposure and underwent class switching and clonal expansion.

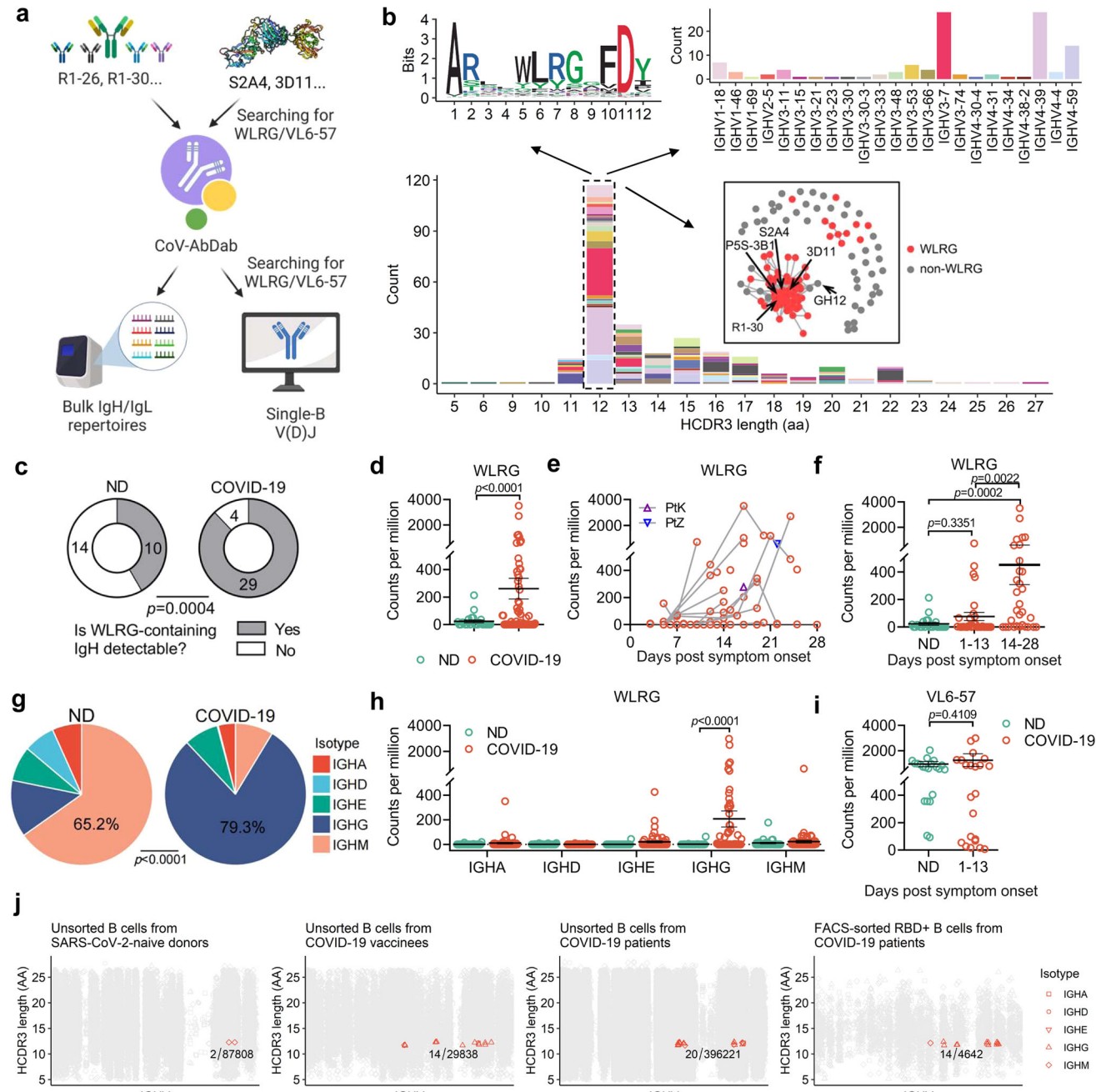

**Fig. 4 | Analysis of VL6-57 light-chain sequences and associated heavy-chain sequences in antibody repertoires. a** A schematic diagram showing the search workflow for VL6-57 light chain and the associated "WLRG" motif containing heavy-chain sequences from CoV-AbDab or published antibody repertoire datasets. Figure 4a, created with BioRender.com, released under a Creative Commons Attribution-NonCommercial-NoDerivs 4.0 International license. **b** Heavy-chain sequence analysis of the 290 VL6-57 utilizing mAbs identified from the CoV-AbDab. A histogram showing the HCDR3 length distribution among the 290 mAbs (bottom). Sequence logo plot showing consensus HCDR3 amino acids of the VL6-57 utilizing mAbs with 12-AA long HCDR3s (top left). A histogram showing the heavy-chain germline gene usage among the VL6-57 utilizing mAbs with 12-AA long HCDR3s (top right). Lineage structure of the VL6-57 utilizing mAbs with 12-AA long HCDR3s based on similarity of HCDR3 sequences (boxed). When the similarity of any two mAbs' HCDR3s is ≥80%, they are linked by a line. mAbs containing the HCDR3 "WLRG" motif are colored red. **c** Pie chart showing the occurrence of "WLRG" motif containing IgH sequences in a cohort of 33 COVID-19 convalescents (sample size, $n = 63$) and 24 naive donors (ND, $n = 24$). **d** Read-count comparison of "WLRG" motif containing IgH sequences in the IgH repertoires of COVID-19

convalescents ($n = 63$) and naive donors ($n = 24$). **e** Read-count dynamics of "WLRG" motif containing IgH sequences in the IgH repertoires of COVID-19 convalescents, each line connects data of different timepoints from the same patient. **f** Read-count comparison of "WLRG" motif containing IgH sequences in the IgH repertoires of COVID-19 convalescents and naive donors across different timepoints ($n = 24, 32, 31$ for ND, 1-13, and 14-28 group, respectively). **g** Pie chart showing the isotype distribution of "WLRG" motif containing IgH sequences in COVID-19 patients and naive donors. **h** Read-count comparison of "WLRG" motif containing IgH sequences between COVID-19 convalescents ($n = 63$) and naive donors ($n = 24$) across different isotypes. **i** Read-count comparison of "QSYDSS" containing VL6-57 light-chain sequences in the IgL repertoires of COVID-19 convalescents and naive donors ($n = 22, 23$ for ND and 1-13 group, respectively). Antibody sequence analyses in **c**–**i** used published bulk sequencing datasets[13,49,50]. **j** The occurrence of VL6-57 mAbs with paired heavy chains containing the HCDR3 "WLRG" motif in the single-B V(D)J repertoires of COVID-19 patients, vaccinees and naive donors[51–56]. Data are presented as mean values ± SEM in **d**, **f**, **h**, and **i**. The two-sided chi-square test was performed in **c** and **g**. Two-tailed unpaired student's *t* test was performed in **d**, **f**, **h**, and **i**. Source data for **d**–**f**, **h**, and **i** are provided as a Source Data file.

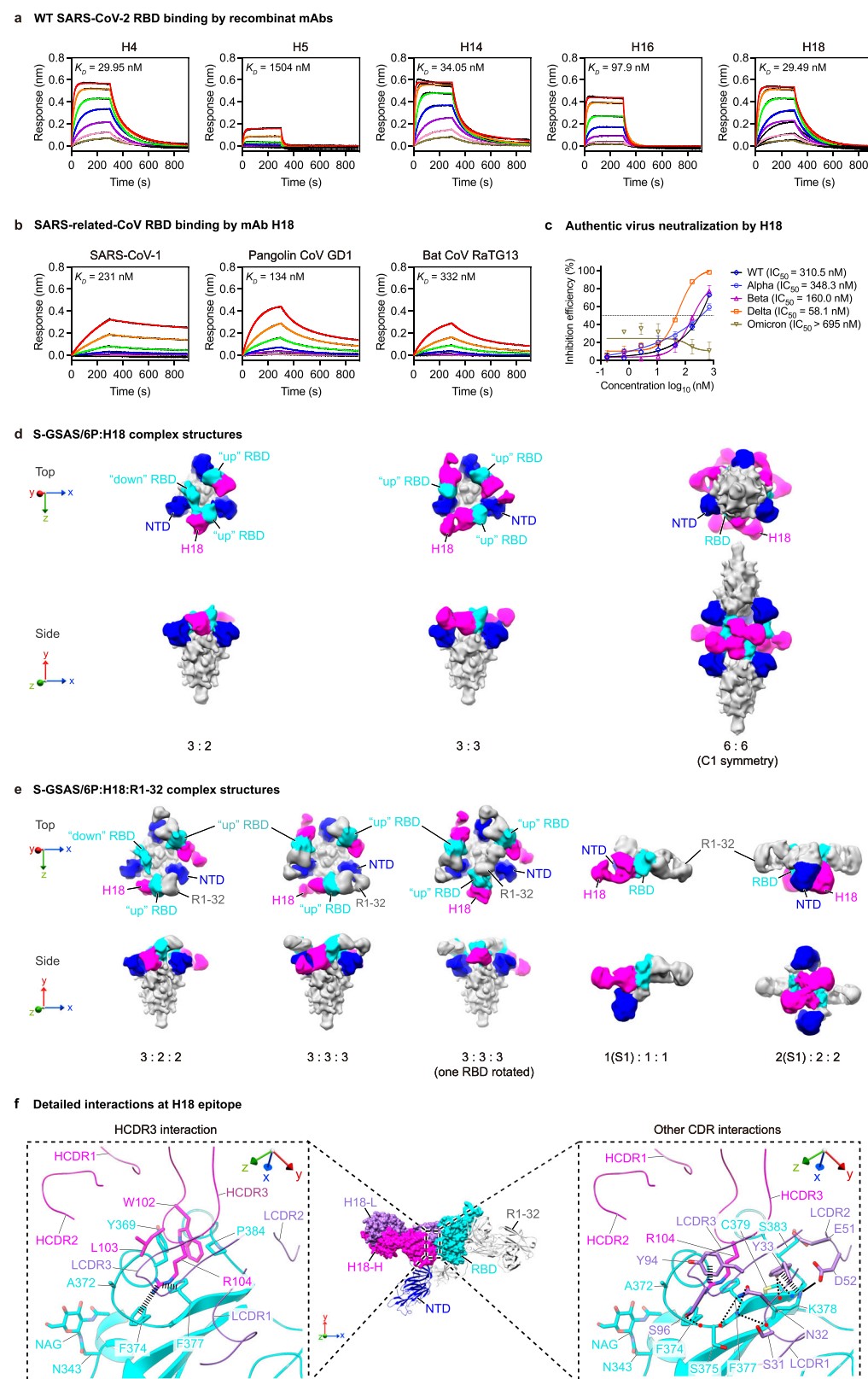

**a** WT SARS-CoV-2 RBD binding by recombinant mAbs

**b** SARS-related-CoV RBD binding by mAb H18

**c** Authentic virus neutralization by H18

WT (IC50 = 310.5 nM)
Alpha (IC50 = 348.3 nM)
Beta (IC50 = 160.0 nM)
Delta (IC50 = 58.1 nM)
Omicron (IC50 > 695 nM)

**d** S-GSAS/6P:H18 complex structures

**e** S-GSAS/6P:H18:R1-32 complex structures

**f** Detailed interactions at H18 epitope

## Antibodies utilizing VL6-57 light chains identified from SARS-CoV-2 naive individuals bind to SARS-CoV-2 RBD

To confirm whether the VL6-57 light-chain (IgL) sequences identified from bulk antibody repertoires of SARS-CoV-2-naive individuals (Fig. 4c) are able to react with SARS-CoV-2 RBD, five different "WLRG" motif-containing heavy chains encoded by IGHV3-7, IGHV3-11, IGHV4-31, IGHV4-39 and IGHV4-61 respectively from 5 different naive donors[13,49,50] are selected to pair with a common VL6-57 light chain shared by the 5 naive donors to generate 5 recombinant mAbs (namely H4, H5, H14, H16 and H18) (Fig. S10a–c). The selected heavy chains are predicted to bind SARS-CoV-2 RBD based on their HCDR3 sequences. All these heavy chains have none or only a few somatic mutations (Fig. S10a–c). BLI assays confirmed that all of the 5 recombinant mAbs are able to bind to wildtype SARS-CoV-2 RBD with affinities ($K_D$) in the range of 30-1504 nM (Fig. 5a,

**Fig. 5 | Characteristics of recombinant mAbs derived from SARS-CoV-2 naive donors. a** WT SARS-CoV-2 RBD binding by five recombinant mAbs were assessed by BLI. Black and colored lines represent experimentally recorded sensorgram traces and corresponding fits. Twofold serially diluted RBD solutions (200 nM to 3.125 nM) were used in the BLI assays. **b** Binding of mAb H18 to SARS-CoV-1, Pangolin CoV GD1, and Bat CoV RaTG13 RBDs were measured by BLI. Kinetic parameters for **a** and **b** are summarized in Supplementary Table 3. **c** Neutralization activities (data are presented as mean values ± SD, *n* = 3) of mAb H18 towards SARS-CoV-2 authentic viruses in cell culture. **d** Structures of S-GSAS/6 P S-trimers in complex with H18 Fabs in different stoichiometries and conformations. **e** Structures of S-GSAS/6 P S-trimers or S1s in complex with H18 Fabs and R1-32 Fabs. Structures in **d**, **e** are low-pass filtered to 12 Å to reveal flexible regions (also see Fig. S2). H18 Fab, NTD, and RBD are highlighted in magenta, blue, cyan, respectively; other structures are colored gray. **f** Detailed H18 epitope structure. H18-H and H18-L chains are colored in magenta and purple; CDR loops are indicated, selected interacting residues between RBD and H18 are shown and indicated; thick and thin dashed lines indicate cation-π interactions and hydrogen bonds. Source data for **c** are provided as a Source Data file.

Supplementary Table 3). Non-VL6-57 light chains of S309 (VK3-20) and CR3022 (VK4-1) were also used to pair with the selected heavy chains to generate recombinant mAbs as negative controls. Except for the pairing between H18-H & CR3022-L, 9 recombinant control mAbs were successfully expressed (Fig. S10d). As expected, ELISA shows that these 9 mAbs are unable to bind S-RBD (Fig. S10e). Notably, H18 exhibits detectable cross-reactivities with RBDs of SARS-CoV-1, Bat CoV RaTG13, and Pangolin CoV GD1, with affinities ranging between 134–332 nM (Fig. 5b, S11, Supplementary Table 3). H18 has mild neutralization activity towards wildtype, Alpha, Beta and Delta SARS-CoV-2 authentic viruses (Fig. 5c), it also possesses activity to trigger conformational change of SARS-CoV-2 S and exhibits inhibition activity in the S-protein-ACE2 interaction mediated cell-cell membrane fusion assay (Fig. 2c, d).

To confirm whether H18 recognizes the same epitope with other VL6-57 mAbs, we obtained cryo-EM structures of wildtype SARS-CoV-2 S-trimer bound to H18 Fab (Figs. 5d, S1b, S2a–c). Similar to cryo-EM structures of SARS-CoV-2 S-trimer:R1-26 complexes (Fig. 2a), we observed 3:2, 3:3 (S-protomer:Fab) SARS-CoV-2 S-trimer:H18 complexes and head-to-head aggregate of two S-trimers each bound by 3 H18 Fabs. Unfortunately, we were not able to uncover high-resolution information on the H18 epitope from these structures. To characterize fine characteristics of the H18 epitope, we generated a ternary complex sample consisting of wildtype SARS-CoV-2 S-trimer, H18 Fab and R1-32 Fab[16] (Figs. 5e, S1c, S2a–c). In the ternary complex sample, most S-trimer particles are bound by three H18 Fabs and three R1-32 Fabs in a 3-RBD "up" conformation (Figs. 5e, S1c). Although the cryo-EM sample was generated using a H18 concentration the same as that of R1-26 in the S-GSAS/6 P:R1-26 sample, we observed a greater degree of S-trimer disassembly likely due to ternary complex formation by the addition of extra R1-32 Fab, structures of S1 bound to both H18 and R1-32 Fabs and dimers of S1 bound to both H18 and R1-32 Fabs were observed (Fig. 5e). Similar S-trimer disassembly has been observed previously in the presence of mAb-R1-32[16,56].

A 3.6 Å resolution structure of the H18 binding interface is derived from a focused refinement of the S1:H18:R1-32 Fab complex (Figs. 5f, S2d). Although the heavy-chain genes are different between H18 (VH4-61) and R1-26 (VH3-7), H18 binds the identified convergent epitope with HCDR3 residues W102 and R104 via hydrophobic contact and cation-π interaction respectively. Consistent with our expectation, the R104 residue within the "WLRG" motif is interacting with Y94 within the LCDR3 "QSYDSS" motif via a cation-π interaction. Antigen binding by H18 HCDR3 is further stabilized by hydrogen bonds from LCDR1 similar to R1-26 (Figs. 5f, S2d). These observations confirm that VL6-57 light chain can pair with multiple heavy chains to target the identified convergent epitope on SARS-CoV-2 spike RBD.

### Molecular basis of evasion from the shared VL6-57 antibody response by SARS-CoV-2 Omicron variants

During the structural analysis of VL6-57 antibodies we noticed that residues S371, S373 and S375 are within the convergent epitope in the ancestral SARS-CoV-2 spike RBD (Fig. 3), these residues have been considered as a feature of Omicron variants and shown to alter virological behavior of the virus[28,29]. We also noted that among the 26 RBD substitutions observed in SARS-CoV-2 so far (Fig. 6a), 18 substitution positions are shared among SARS-related-CoVs (Fig. 6b) and

among the other 8 substituted positions, S371, S375, T376, and R408 fall within the epitope of the VL6-57 class 4 public antibodies (Fig. 6a, b). Mapping of substituted RBD positions and VL6-57 antibody epitope areas indicates that the epitope is relatively conserved among SARS-related-CoVs. Although R1-26 and H18 could bind and neutralize SARS-CoV-2 WT and early VOCs, both of them fully lose binding and neutralization abilities towards Omicron BA.1 (Figs. 1d, e, 5c). BLI assays show that simultaneous rescue mutations at 371, 373 and 375 on Omicron BA.1 RBD (Omicron-(L371S + P373S + F375S)) are able to completely recover binding by VL6-57 mAbs R1-26 and H18 (Fig. 6c, Supplementary Table 4). This result indicates that the evasion of VL6-57 mAb neutralization is specifically mediated by substitutions at S371, S373, and S375. BLI assays using a series of RBD mutants revealed that single S371L, S373P or S375F mutation greatly reduces binding by R1-26 and H18. Combinations of any two simultaneous mutations (S373P/S375F, S371L/S375F, S371L/S373P) are able to almost completely abolish R1-26 and H18 binding (Fig. 6c, Supplementary Table 4). Structural comparison of WT, Omicron BA.1 and BA.2 RBDs reveals that backbone conformations at residues 373 to 375 are affected by the S373P introduced proline and the phenylalanine sidechain introduced by S375F (Fig. 6d), these epitope structural changes likely abolish binding by the VL6-57 mAbs.

## Discussion

We identify a class of VL6-57 light chain utilizing antibodies with the ability to pair with diverse heavy chains to target a convergent epitope defined by featured Omicron mutations - S371L/F, S373P, S375F. The identified epitope is cryptic in a "down" RBD and strongly hydrophobic, consistent with that HCDR3s of VL6-57 mAbs share hydrophobic residues to interact with the epitope. Notably, within the epitope, residues S371-S373-S375 are located at the entrance to the identified fatty acid binding pocket of sarbecovirus spike RBD. Binding of linoleic acid within the pocket has been associated with "locked" spike conformation with potential functions in virus assembly[36–38,57,58]. Being a highly dynamic protein, SARS-CoV-2 S-protein has been observed to adopt RBD "up" and "down" conformations. D614G substitution became fixed in the spike protein shortly after the SARS-CoV-2 pandemic[36,59–61], it has been shown that this substitution increased spike stability and a shift towards a more open (RBD "up") S-trimer has been observed by multiple studies[36,59,62]. We speculate that the change in spike dynamics may have increased immune pressure posed by mAbs targeting cryptic epitopes. Indeed, we find that binding of R1-26, H18 and possibly other VL6-57 mAbs to the convergent cryptic epitope not only blocks ACE2 binding[41], but also prematurely triggers spike conformational change resulting in spike inactivation. When paired with other RBD-specific antibodies, binding of antibodies to cryptic epitopes may also result in spike inactivation by promoting S-trimer disassembly[16,56]. The increased immune pressure may promote the introduction of S371L/F-S373P-S375F in Omicron variants. The S371L/F-S373P-S375F mutations have also been found to stabilize the interaction between up-down RBDs within the Omicron S-trimer, likely further modulating S-trimer dynamics, stability and receptor binding[56,63–66]. The identified cryptic epitope of VL6-57 mAbs is relatively conserved among S-proteins of sarbecoviruses[26]. Before the emergence of Omicron variants, the VL6-57 antibody epitope remains

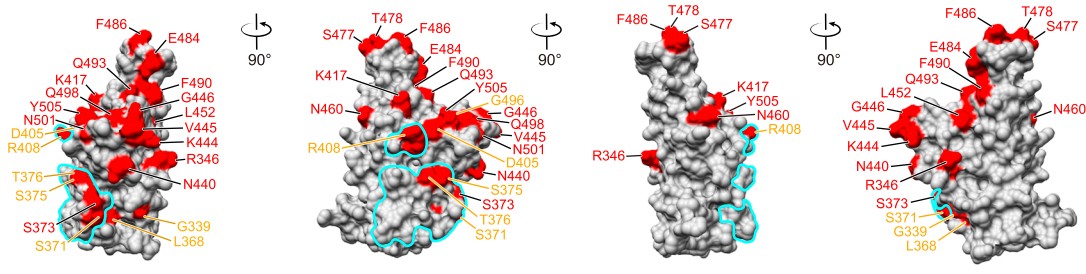

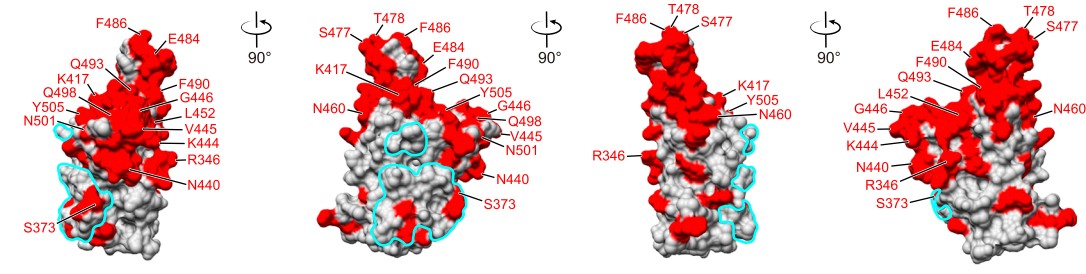

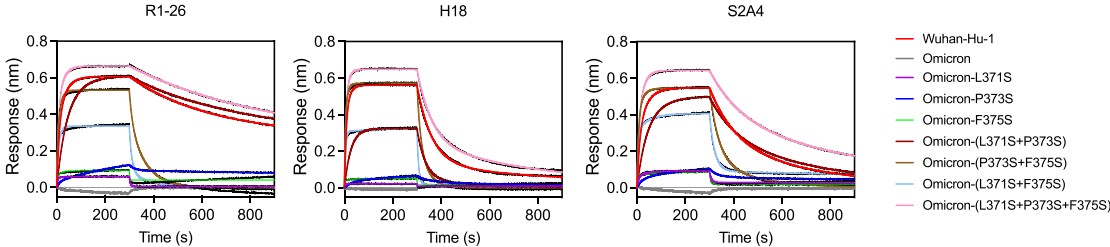

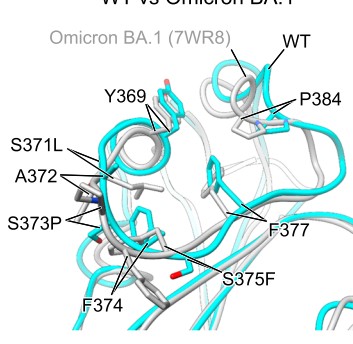

**Fig. 6 | SARS-CoV-2 Omicron variants evade VL6-57 antibodies by mutations at residues S371-S373-S375.** a Substituted residues among SARS-CoV-2 variants are highlighted in red on the surface of SARS-CoV-2 RBD. Among the 26 substitutions, 18 are shared among SARS-related-CoVs and are labeled in red, the other 8 substitutions are labeled in orange. **b** Substituted residues among RBDs of SARS-related-CoVs are highlighted in red. Among the 72 substitutions, 18 are shared with SARS-CoV-2 variants and are labeled in red. Epitope of R1-26 on RBD is encircled by cyan outlines in **a** and **b**. **c** Binding of R1-26, H18, and S2A4 to a series of Omicron BA.1 RBD mutants carrying 1-3 rescue mutations at S371, S373, and S375. BLI binding kinetic parameters are summarized in Supplementary Table 4. **d** Comparison of the VL6-57 mAb epitope structures on WT and Omicron BA.1 (left) or BA.2 (right) RBDs. Superposition shows differences in amino-acid sidechain orientations and main chain backbone structures. Source data for **a** and **b** are provided as a Source Data file.

unchanged in early SARS-CoV-2 VOCs or VOIs. We and others also found that some of the VL6-57 mAbs including H18 and many others[67] are able to cross-react with SARS-CoV-1 RBD, confirming that the VL6-57 epitope is relatively conserved among sarbecoviruses and suggesting that cross-reactive VL6-57 mAbs may be used to combat potential emergent sarbecoviruses. On the other hand, the

introduction of S371L/F-S373P-S375F demonstrates an extraordinary adaptability of SARS-CoV-2, providing a piece of evidence to suggest that unusual changes of a pandemic virus may occur in response to potential strong population immune pressures including those posed by population antibodies. Studies have shown that single mutation of S371F/L, S373P or S375F strongly impairs spike processing and spike

mediated virus entry[28]. Among them, S375F has the most detrimental effect[29], and notably, mutation at this position has not been observed among animal sarbecovirus S-proteins[26] (Fig. 6b).

The above observations highlight the likely necessity of the S371-S373-S375 mutations for SARS-CoV-2 Omicron variants, even at the expense of drastic S-protein phenotypic change with associated impact on virus behavior. It has been shown that antigenic changes can alter SARS-CoV-2 receptor binding[68], for other viruses, antigenic evolution of the pandemic H3N2 virus was found to substantially weaken hemagglutinin (HA) sialic acid binding affecting virological phenotypes[69]. While our study provides evidence to show that the S371-S373-S375 mutations enable SARS-CoV-2 to evade the identified class of VL6-57 public antibodies, other functions for these mutations continue to emerge. A recent study reports that S375F promotes immune evasion by enhancing spike binding to Siglec-9 on macrophages to impair antigen phagocytosis and presentation[30]. A recent preprint also shows that when G339H, S371F, S373P, and S375F mutations are reverted in a reverse-genetic authentic virus carrying Omicron BA.1 S-protein in an ancestral virus genetic background, the Omicron nasal epithelium cell tropism is almost completely abrogated[27]. However, a complex epistatic relationship between S-protein mutations has been suggested—S371L/F, S373P and S375F may only be adaptive when they co-occur with the other Omicron-specific S-protein substitutions[26,27]. Indeed, the correlation between SARS-CoV-2 tissue tropism, pathogenicity, immune evasion, and transmissibility is highly complex and remains ill-defined. Other factors affecting SARS-CoV-2 infection continue to emerge, utilization of membrane-type matrix metalloproteinases (MT-MMPs) and ADAM (a disintegrin and metalloproteinase) has recently been proposed to correlate with nasal cell tropism[70–72]. Other Omicron tissue tropism and pathogenesis determinants related to host interferon responses have been recently implicated[72–78]. Some Omicron attenuation determinants have been mapped outside of the S-protein gene[79,80]. Nevertheless, multiple functions identified so far for the S371L/F, S373P and S375F mutations illustrate vividly that antigenic change of viral surface protein can have complex consequences and a delicate balance among tissue tropism, pathogenicity, immune evasion, and transmissibility is likely pursued by circulating viruses[1].

It has been reported that SARS-CoV-2 infections induce several classes of germline antibodies of specific VH genes. Due to similarity of equivalent germline antibody genes among humans, induced classes of germline antibodies are widely present in the population. Certain germline antibodies of the same VH gene can bind a convergent spike epitope with similar molecular interactions mediated by antibody residues encoded by germline sequences[45,49,81,82]. Similar phenomena have also been observed for mAbs induced by infections of other viruses[83–85]. A common feature of these germline antibodies is that essential paratope interactions are primarily mediated by antibody heavy chains, such feature has been observed for VH3-53[14], VH1-58[86], VH1-2[19,20], VH1-69[16], and VH1-24[87] SARS-CoV-2 S-specific antibodies. These heavy-chain mediated shared antibody responses have been thought to be a key driving force of SARS-CoV-2 antigenic drift[18,88,89]. In this study, by contrast, the identified population antibody response is mediated by antibodies with shared usage of VL6-57 light-chain gene. Generally, heavy chains dominate interactions in antibody-antigen complexes, while light chains help modulate heavy-chain conformation[90]. It has been shown that antibodies of VH1-69 heavy chains can pair with irrelevant light chains to bind influenza HAs[91,92]. Our structural study reveals that interactions between the VL6-57-encoded mAbs and RBD are mainly mediated by germline gene encoded residues in HCDR3 and LCDR1, while germline gene encoded LCDR3 residues support HCDR3 in antigen binding. Although VL6-57 is most frequently observed to pair with VH3-7, VH4-39, and VH4-59 we provide evidence that many other heavy chains can also pair with VL6-57 to bind SARS-CoV-2 RBD. Therefore, VL6-57 germline sequence

provides an efficient framework to allow pairing with diverse heavy chains to generate SARS-CoV-2 RBD-targeting antibodies, such an unique property of VL6-57 gene increases the probability of VL6-57 class of public antibodies being induced across a broad population. This study provides a rare example that a light-chain paired with common binding motifs in various heavy chains can largely dictate the binding mode of a class of antiviral antibodies and provides a molecular mechanism to explain the increased identification of VL6-57 S-specific mAbs among SARS-CoV-2 infected individuals[45,47,48].

Given the epitope of the VL6-57 mAbs is largely conserved among animal sarbecoviruses, we speculate that the versatility of VL6-57 in generating SARS-CoV-2 RBD antibodies may be unique within the human population. Wide induction of this class of antibodies contribute to selection pressure on SARS-CoV-2 within the human population, but similar response may be absent in other species. Therefore, S371L/F-S373P-S375F represents a specific constellation of immune escape mutations for SARS-CoV-2 adaptation and transmission within the human population. However, it is worth noting that there is still VL6-57 mAb that is resistant to Omicron variants[93], implying adaptation of antibody immunity to viral mutations. Therefore, exploring the co-evolution of virus and antibody immunity in future works should be beneficial to understanding co-adaptability between virus and host[94]. Overall, this study provides a fresh example of light-chain mediated populational antibody immune pressure against SARS-CoV-2 at molecular level. Our findings lend support to the hypothesis that convergent antibody responses may influence viral antigenic drift. However, establishing a direct causal link between immune pressure and the emergence of specific mutations such as those seen in Omicron variants likely requires additional longitudinal and functional studies. Also, our findings highlight the importance of monitoring viral sequences for mutations in key epitopic regions. Future vaccine strategies could benefit from incorporating such surveillance data to pre-emptively adjust vaccine compositions in response to emerging escape variants, ensuring continued vaccine effectiveness.

## Methods
### Cells and viruses
Expi293F cells (Thermo Fisher Scientific, A14527) were maintained in Expi293F Expression medium (Thermo Fisher Scientific, A1435101) at 37 °C by shaking at 120 rpm under a humidified atmosphere with 8% $CO_2$. The human embryonic kidney (HEK) 293T (ATCC, CRL-3216) and Vero E6 (ATCC, CRL-1586) were maintained in Dulbecco's Modified Eagle's Medium (DMEM) supplemented with 10% FBS. The authentic SARS-CoV-2 viruses, including WT (Wuhan-Hu-1), Alpha (B.1.1.7), Beta (B.1.351), Delta (B.1.617.2), and Omicron (BA.1) were isolated from COVID-19 patients and preserved in Guangzhou Customs District Technology Center BSL-3 Laboratory. The SARS-CoV-2 Delta strain was a gift from Guangdong Provincial Center for Disease Control and Prevention, China. Experiments related to authentic SARS-CoV-2 viruses were conducted in Guangzhou Customs District Technology Center BSL-3 Laboratory.

### Expression of monoclonal antibody
The antibody heavy- and light-V genes (VH/VL) were cloned into human IgG1 expression vectors using Clone Express II One Step Cloning Kit (Vazyme, China, Cat# C112-02). When the density of HEK293F cells reached $1 \times 10^6$ cells/mL, equal amounts of heavy- and light-chain plasmids were transfected into HEK293F cells using EZ cell transfection reagent (Life-iLab Biotech, China, Cat# AC04L092). Following transfection, HEK293F cells were cultured in CD 293 TGE medium (ACRO, China, Cat# CM-1156) containing 10% CD Feed X supplement (ACRO, China, Cat# CF-1116-12) at 37 °C in a humidified atmosphere with 5% $CO_2$ and shaking at 120 rpm. 6 days post transfection, supernatants were harvested and clarified by centrifugation. Supernatants were filtered through 0.22–μm filters (Merck Millipore,

Germany, Cat# SLGP033NS) before incubation with Protein A Resin (Genscript, China, Cat# L00210) at room temperature for 2 h for antibody affinity purification. After washing, antibodies were eluted from the Protein A Resin using 0.1 M Na–citrate (pH 3.25) and eluents were neutralized immediately with 1 M Tris-HCl (pH 8.8). Antibodies were concentrated in PBS using Amicon Ultrafilter devices (Merck Millipore, USA, Cat# UFC810096) and stored at −80 °C.

## Inhibition of SARS-CoV-2 S-mediated cell-cell fusion

As previously described[44], we utilized a real-time, quantitative live cell split-GFP fluorescence complementation phenotypic assay to analyze the fusion inhibition activity of antibodies. Briefly, effector cells (HEK293T cells expressing SARS-CoV-2 S and GFP1-10 proteins) were seeded into a 96-well-plate ($5 \times 10^4$ cells per well in DMEM). Each well was cultured in the presence or absence of a test mAb at the indicated concentrations for 2 h at 37 °C. Subsequently, target cells (hACE2-293T cells expressing GFP11 protein, $5 \times 10^4$ cells per well in DMEM) were added uniformly to each well, and fluorescence images were taken 2 h after the addition of target cells using a Nikon fluorescence microscope or a Keyence BZ-X800 all-in-one fluorescence microscope. The GFP area in each well was quantified on Image J, and percentage inhibition of cell-cell fusion was calculated using the following formula: $(1 − (E − N)/(P − N)) \times 100\%$. "E" represents the GFP area in the experimental group. "P" represents the GFP area in the positive control group, where no mAb was added. "N" is the GFP area in the negative control group, where effector HEK293T cells only express GFP1-10. Samples were tested in triplicate, and experiments were repeated at least twice.

## Protein expression and purification

The extracellular domain of SARS-CoV-2 spike (S) protein (residues 14-1211) with an N-terminal mu-phosphatase signal peptide, a "R" substitution at the multibasic furin cleavage site (R682-R685), and a C-terminal TEV protease cleavage site followed by a T4 fibritin trimerization motif and a His$_6$ tag was cloned into the mammalian expression vector pCDNA3.1(+), named "S-R", as previously described[43]. To generate the stabilized S-protein for cryo-EM sample preparation, the sequence of S-protein was modified with six prolines at residues 817, 892, 899, 942, 986, and 987 and the furin cleavage site was changed to "GSAS"[95], named "S-GSAS/6 P". To express the SARS-CoV-2 RBD, residues 319−541 of S-protein were fused with an N-terminal mu-phosphatase signal peptide and a C-terminal 6×His tag before the sequence was inserted into the pCDNA3.1(+) vector. S-protein or RBD expression vector was transiently transfected into Expi293F using polyethylenimine. Proteins were purified using IMAC (immobilized metal affinity chromatography) following previously described protocols[16,37,56]. All proteins were aliquoted, flash-frozen in liquid nitrogen and stored at −80 °C.

## Biolayer interferometry

Binding assays were carried out on an Octet RED96 instrument (Sartorius) using a previously established protocol[16]. Briefly, each Protein A biosensors (Sartorius) was pre-equilibrated in PBST buffer (PBS, pH 7.4, 0.02% Tween-20, 1 mg/ml bovine serum albumin) for 10 min. Subsequently, IgG at 11 µg/ml was loaded onto each biosensor to a response level between 1.6-1.8 nm. The IgG immobilized biosensors were submerged into twofold serially diluted (200-3.125 nM) RBD or S-protein solutions for 300 s to record association. The biosensors were subsequently submerged into PBST buffer for 600 s to record dissociation. For IgG binding to the generated RBD mutants, biosensors immobilized with IgGs were monitored for association (300 s) in RBD solutions at a fixed concentration of 200 nM before the sensors were submerged into PBST to monitor dissociation (600 s). IgG-immobilized sensors were also submerged into a PBST buffer to record references. Data were reference-subtracted and analyzed using Data

Analysis HT v12.0.2.59 software (Sartorius) with a 1:1 fitting model for binding to RBDs and 2:1 fitting model for binding to S-trimers. Raw data and fits were plotted in GraphPad Prism v8.0.

Competition assays were performed on a Gator label-free bioanalysis system (GatorBio). 2 µg/ml of SARS-CoV-2 RBD (Sino Biological, Cat# 40592-V08B) was immobilized onto the pre-equilibrated Anti-His biosensors (GatorBio). Biosensors were statured with the first antibody for 300 s before submerging into the second antibody or ACE2 solutions for 200 s. Data were analyzed by the Gator data analysis software (GatorBio) and plotted in GraphPad Prism v8.0.

## Ligand-induced conformational change of spike protein

S-R diluted to 1 mg/ml (7.09 µM) was incubated with ACE2-Fc or antibodies in IgG or Fab form at a 1:1.1 molar ratio at room temperature for 1 h. The samples were subsequently treated with 50 µg/ml proteinase K at 4 °C for 30 min. Each sample was boiled in 5 × non-reducing SDS loading buffer at 98 °C for 5 min to stop the reaction. Samples were separated by SDS-PAGE on 4-12% gradient gels, before protein bands were transferred onto a polyvinylidene difluoride membrane using a semi-dry blotting system. The membrane was blocked with 5% milk in PBST before the membrane was incubated with a primary antibody (rabbit anti-SARS-CoV-2 S2 polyclonal antibody, Sino Biological, Cat# 40590-T62, 1:2500 dilution) in PBST. After extensive washing, the membrane was incubated with the secondary antibody (horseradish peroxidase-conjugated goat anti-rabbit IgG, Beyotime, Cat# A0208, 1:2500 dilution) in PBST. Finally, protein blots were visualized by chemiluminescence using a Pierce ECL Western Blotting Substrate (Thermo Fisher Scientific, Cat# 32106).

## Cryo-EM sample preparation and data collection

To generate S-GSAS/6 P:R1-26 Fab, S-GSAS/6 P:H18 Fab complexes, S-GSAS/6 P at 4.43 mg/ml was incubated with R1-26 Fab or H18 Fab at a 1:1 molar ratio. To generate S-GSAS/6 P:H18 Fab:R1-32 Fab complex, S-GSAS/6 P at 4.43 mg/ml was incubated with H18 and R1-32 Fabs at a 1:1:1 molar ratio. After 1 min incubation at room temperature, each 3 µl sample was supplemented with 0.1% octyl-glucoside (Sigma-Aldrich, Cat# V900365) before it was applied onto a 300-mesh holey carbon-coated copper grid (Quantifoil, Cu R1.2/R1.3) pre-treated by glow-discharging at 15 mA for 30 s. Each grid was blotted for 2.5 s with a blot force of 4 at 22 °C and 100% humidity before plunge-freezing in liquid ethane using a Vitrobot Mark IV (Thermo Fisher Scientific). The S-GSAS/6 P:R1-26 complex cryo-grid was imaged in a Titan Krios electron microscope (Thermo Fisher Scientific) operating at 300 kV and equipped with Gatan BioQuantum energy filter (slit width 20 eV) and Post-GIF Gatan K3 Summit direct detection camera. Movie stacks were automatically recorded using EPU at a nominal magnification of ×81,000 in super-resolution mode with a calibrated pixel size of 0.5475 Å and nominal defocus values ranged between −0.8 to −2.0 µm. Each stack was fractionated into 38 frames and exposed at a dose rate of 25 e⁻/pixel/s for 2.4 s resulting in a total dose of ~ 50 e⁻/Å². The S-GSAS/6 P:H18 complex cryo-grid was imaged in a Talos Arctica electron microscope (Thermo Fisher Scientific) operating at 200 kV. Using the SerialEM v3.8.7 software, movie stacks were recorded at a nominal magnification of ×45,000 on a K3 direct detection camera (Gatan) in super-resolution mode with a calibrated pixel size of 0.44 Å with nominal defocus values ranged between −0.8 to −2.5 µm. Each stack was fractionated into 27 frames and exposed at a dose rate of 24.4 e⁻/pixel/s for 1.89 s, resulting in a total dose of ~60 e⁻/Å². The S-GSAS/6 P:H18:R1-32 complex cryo-grid was imaged in a 300 kV Titan Krios electron microscope (Thermo Fisher Scientific) equipped with a SelectrisX energy filter (slit width 10 eV) and a Falcon 4 direct electron detector. Movie stacks were automated collected using EPU software with the electron event representation (EER) mode at a nominal magnification of ×130,000 with a pixel size of 0.93 Å and nominal defocus values ranged between −0.6 to −2.0 µm. Each stack was

recorded and exposed at a dose rate of 7.51 e⁻/pixel/s for 5.79 s resulting a total dose of ~50 e⁻/Å². These settings yielded EER stacks consisting of 199 frames. All movie stacks were imported into cryoS-PARC live (v3.3.2)[96] for pre-processing, which includes patched motion correction, contrast transfer function (CTF) estimation and bad images rejection. Movie stacks of S-GSAS/6 P:R1-26 complex and S-GSAS/6 P:H18 complex datasets were binned 2× resulting in pixel sizes of 1.095 Å and 0.88 Å, respectively.

## Cryo-EM data processing

Data processing was carried out using cryoSPARC v3.3.2/v4.2.0. After bad image removal, particles were picked by blob-picking on 4343 S-GSAS/6 P:R1-26, 4822 S-GSAS/6 P:H18 and 9861 S-GSAS/6 P:H18:R1-32 images. After 2D Classification, several good 2D classes were selected as the templates for template-picking, resulting in initial datasets of 2,359,986 particles for S-GSAS/6 P:R1-26; 2,357,740 particles for S-GSAS/6 P:H18; and 3,335,676 particles for S-GSAS/6 P:H18:R1-32. Template-picked particles were extracted for two or more rounds of 2D Classification to remove contaminants and low-quality particles. For S-GSAS/6 P:H18:R1-32, particles with intact S-trimers were re-extracted with a larger box size before the last round of 2D Classification. After several rounds of 2D Classification, well-aligned particles with intact S-protein features were selected and subjected to Ab-initio Reconstruction to generate 3D models. Particles of initial models with complete S2 but incomplete S1 were further classified using Ab-initio Reconstruction with class similarity value 0.3 for the S-GSAS/6 P:R1-26 dataset, 0.7 for the S-GSAS/6 P:H18 and S-GSAS/6 P:H18:R1-32 datasets to separate different conformations.

3:2 and 3:3 S-protomer:Fab structures were observed in both S-GSAS/6 P:R1-26 and S-GSAS/6 P:H18 datasets. Head-to-head aggregates of 2 S-trimers bound by 3 Fabs, giving S-protomer:Fab stoichiometry of 6:6, were observed in 2D Classifications and Ab-Initio Reconstructions of both S-GSAS/6 P:R1-26 and S-GSAS/6 P:H18 datasets. Particles of aggregate classes were selected and combined before particles were re-extracted with a larger box size. A S-trimer aggregate structure from the S-GSAS/6 P:R1-26 dataset was obtained by a round of 2D Classification followed by Ab-Initio Reconstruction. Similarly, 2 S-trimer aggregate structures were obtained from the S-GSAS/6 P:H18 dataset. To obtain a higher resolution structure of the interface between RBD and R1-26 Fab, particles of the 3:3 S-GSAS/6 P:R1-26 structure were refined again with applied C3 symmetry by Non-uniform Refinement. Refined particles were subjected to Symmetry Expansion and Density Subtraction before a Local Refinement was carried out focusing on a region containing RBD and Fab-V.

For the S-GSAS/6 P:H18:R1-32 dataset, one 3:2:2 and two 3:3:3 structures were observed. In the S-GSAS/6 P:H18:R1-32 dataset, many well-defined 2D classes showing incomplete S-protein features were selected for additional Ab-Initio Reconstruction. One 1:1:1 S1:H18:R1-32 structure and one 2:2:2 S1:H18:R1-32 structure consisting of two head-to-head 1:1:1 S1:H18:R1-32 complexes were observed. Particles of these two structures were re-extracted with a smaller box size. Final maps of differently conformations were reconstructed using Non-uniform Refinement.

All maps with resolutions higher than 4 Å were processed by a second round of Non-uniform Refinement with defocus refinement and global CTF refinement to improve map quality. All resolutions were estimated at the 0.143 criterion of the phase-randomization-corrected Fourier shell correlation (FSC) curve calculated between two independently refined half-maps multiplied by a soft-edged solvent mask in RELION v4.0[96]. Additional data processing details are summarized in Figs. S1 and S2, and Supplementary Table 2.

## Model building and analysis

A previously determined structure of SARS-CoV-2 S-trimer in complex with 3 R1-32 Fabs and 3 ACE2s (PDB: 7YEG)[16] or a structure of SARS-CoV-2 S1 in complex with a YB9-258 Fab and an R1-32 Fab (PDB: 8HC5)[56] was used as starting model. An antibody light chain (PDB: 7D6I) was used as the starting model for R1-26 and H18 Fab light chains. Starting models for R1-26 and H18 Fab heavy chains were generated from heavy-chain structures in PDBs - 5X8M and 7VSU respectively. Starting models were fitted into final refined maps in UCSF Chimera v1.14[97]. Iterative model building and real space refinement were carried out in Coot v0.9.6[98] and PHENIX v1.20.1-4487[99]. Model refinement statistics are summarized in Supplementary Table 2. Interface analyses were performed in QtPISA v2.1.0[100]. Structure figures were generated in UCSF Chimera v1.14.

## SARS-CoV-2 authentic virus neutralization assay

Antibodies were serial diluted with DMEM and mixed with 200 focus forming unit (FFU) Wuhan-Hu-1 (wildtype), Alpha, Beta, Delta, or Omicron BA.1 authentic SARS-CoV-2 viruses. After incubation at 37 °C for 1 h, antibody-virus mixtures were added to a 96-well plate cultured with Vero E6 cells and incubated at 37 °C in 5% CO₂ for 1 h. After removing the inoculum, each well was overlaid with 100 μL 1.6% car-boxymethylcellulose warmed to 37 °C. After culturing for 24 h, over-lays were removed and the cells were fixed with 4% paraformaldehyde (Biosharp, China, Cat# BL539A) and permeabilized with 0.2% Triton X-100 (Sigma, USA, Cat# T8787). Cells were incubated with a human anti-SARS-CoV-2 nucleocapsid protein monoclonal antibody (obtained by laboratory screening) at 37 °C for 1 h. After washing with 0.15% PBST three times, cells were incubated with an HRP-labeled goat anti-human secondary antibody (Jackson ImmunoResearch Laboratories, Cat# 609-035-213) at 37 °C for 1 h. Plates were washed with 0.15% PBST three times before the foci were visualized by TrueBlue Peroxidase Substrate (KPL, Gaithersburg, MD, Cat# 50-78-02), and counted with an ELISPOT reader (Cellular Technology Ltd. Cleveland, OH). The foci reduction neutralization test titer (FRNT50) was calculated by the Spearman-Karber method.

## Analysis of the structurally characterized SARS-CoV-2 S-specific mAbs

A total of 376 antibody:RBD/S-protein complex structures were curated from the PDB (https://www.rcsb.org/) before May 18, 2023. LCDR2 loop structures of 2-7 (PDB 7T3M, EMD-25663), 553-15 (PDB 7WO7, EMD-32641) were lightly adjusted in Coot v0.9.6[98] according to the corresponding density maps and refined in PHENIX v1.20.1-4487[99]. Epitope residues, paratope residues, and buried surface area (BSA) were determined or calculated using the PDBe PISA server (https://www.ebi.ac.uk/msd-srv/prot_int/). BSA for each epitope residue is assigned as a feature of a certain antibody and used to construct a feature matrix $M_{AxB}$ for downstream analysis, where A is the number of antibodies and B is the number of features (amino-acid length of spike: 1273). Therefore, a 376 × 1273 BSA matrix was obtained, which was subsequently used as input for epitope classification with the R package UMAP (v0.2.9.0). Uniform Manifold Approximation and Projection (UMAP) is an algorithm for dimensional reduction. After the reduction analysis, clustering was performed using the K-means algorithm. UMAP and K-means clusterings were conducted independently. Antibodies clustered into the same cluster as the well-documented mAb CR3022 are defined as class 4 antibodies. After this workflow, 35 class 4 mAbs were identified among the 376 structurally characterized S-specific mAbs (Fig. S5). In addition, we found that class 4 epitope is largely overlapping with the F2 epitope which was previously defined by deep mutational scanning[7]. Thus, a total of 69 F2 epitope mAbs were also included for germline gene usage analysis. Germline gene usages of these structurally characterized S-specific mAbs were inferred using IMGT/V-QUEST (https://www.imgt.org/). Visualization of the germline heavy and light gene usage and pairing among the 35 class 4 and the 69 F2 antibodies were performed in R platform v4.2 using the R package circlize v0.4.10[101].

## Bioinformatic analysis

To investigate features of VL6-57 utilizing mAbs, a total of 290 VL6-57 utilizing mAbs were extracted from the CoV-AbDab database[46] for downstream analysis. Analysis of germline gene usage and CDR3 length were perform using the built-in function in R platform v4.2. CDR3 amino-acid composition analysis was performed using the R package ggseqlogo[102]. VL6-57 utilizing mAbs with 12-AA length HCDR3 were used for lineage structure reconstruction. R package igraph was employed for visualization of the lineage structure of VL6-57 utilizing mAbs with 12-AA length HCDR3s. After the above workflow, the results showed that WLRG motif within 12-AA long HCDR3 and QSYDSS motif within LCDR3 are the convergent signatures of the VL6-57 mAbs. To determine the occurrence of VL6-57 mAbs with paired HCDR3 "WLRG" motif, we analyzed nearly 2 billion IgH and IgL sequences from 3 previously described datasets[13,49,50]. These datasets have been deposited by us in the National Genomics Data Center (https://bigd.big.ac.cn/), China National Center for Bioinformation (CNCB) under accession numbers PRJCA003775, PRJCA007067, and PRJCA017560. HCDR3 "WLRG" motif-containing IgH sequences were defined as those sequences which have 12-AA long HCDR3s with a "WLRG" motif. VL6-57 IgL sequences were defined as those sequences that use the germline gene VL6-57 with 9 to 10-AA long LCDR3s, and encoding a "QSYDSS" motif within LCDR3 region. The abundance of "WLRG" motif-containing IgH sequences or VL6-57 IgL sequences were normalized by counts per million. The clonal expansion of B cells expressing "WLRG" motif-containing IgH or VL6-57 IgL sequences in COVID-19 patients after SARS-CoV-2 infection were tracked. The divergence from germline genes of all IgH or IgL sequences was equal to somatic hypermutations, and sequence identity to queried sequence was calculated using the R package Biostrings v2.60.2 (http://bioconductor.org/packages/release/bioc/html/Biostrings.html). All IgH or IgL sequences were plotted as a function of sequence somatic hypermutations (x axis) and sequence identity (y axis) to heavy or light chains of R1-26, R1-30, R2-3, R2-6, or R2-7 with a color gradient indicating sequence density. To determine the frequency of B cells expressing VL6-57 mAbs with paired HCDR3 "WLRG" motif, we analysis the single-B V(D)J sequences from SARS-CoV-2-exposed and - naive individuals[51–55] and FACS-sorted RBD+ B cells from COVID-19 convalescents[56]. These datasets are available from the Gene Expression Omnibus database under accession numbers: GSE230227, GSE171703, GSE158038, GSE158055; http://www.microbiome-bigdata.com/project/SARS-CoV-2/; and National Genomics Data Center (https://bigd.big.ac.cn/) under the accession number: PRJCA012020. B cells expressing VL6-57 mAbs with paired HCDR3 "WLRG" motif were defined as those B cells expressing heavy chains with a 12-AA long HCDR3 containing a "WLRG" motif and VL6-57 light chains with a "QSYDSS" motif in LCDR3. A schematic diagram summarizing the bioinformatic analysis workflow is shown in Fig. 4a, which is created with BioRender.com, released under a Creative Commons Attribution-NonCommercial-NoDerivs 4.0 International license.

## Reporting summary

Further information on research design is available in the Nature Portfolio Reporting Summary linked to this article.

## Data availability

Cryo-EM density maps for the structures of R1-26 or H18 Fab in complex with S-trimer or S1 fragment have been deposited in the Electron Microscopy Data Bank (EMDB) under accession codes EMD-60099, EMD-60100, EMD-60101, EMD-60102, EMD-60103, EMD-60104, EMD-60105, EMD-60106, EMD-60107, EMD-60108, EMD-60109, EMD-60110, EMD-60111. Related atomic models have been deposited in the Protein Data Bank (PDB) under accession codes 8ZHD, 8ZHE, 8ZHF, 8ZHG, 8ZHH, 8ZHI, 8ZHJ, 8ZHK, 8ZHL, 8ZHM, 8ZHN, 8ZHO, 8ZHP, respectively. Source data are provided with this paper.

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

## Acknowledgements

We thank the staff at cryo-EM Facilities of GIBH-CAS, Guangzhou. Laboratory Bio-imaging Technology Platform and SUSTech for help on cryo-EM sample preparation and data collection. This study was supported by the National Natural Science Foundation of China (82201932 to Q.Y., 92269201 to L.C., 82341085 to X.X., 32170189 and 32241021 to J.H., 82025001 to J.Z.), the National Key Research and Development Program of China (2021YFA1300903 to X.X.), the Emergency Key Program of Guangzhou Laboratory (EKPG21-06 to X.X.), the R&D Program of Guangzhou Laboratory (SRPG22-002 to X.X. and SRPG22-003 to J.H.), the Science and Technology Planning Project of Guangdong Province (2021A1515011289, 2023B1212060050 and 2023B1212120009 to X.X.), China Postdoctoral Science Foundation (2022M710891 to Q.Y.), the State Key Laboratory of Respiratory Disease (SKLRD-Z-202324 to Q.Y.), and the Young Doctoral Starting Sail Project of the Guangzhou Municipal Science and Technology Bureau (2024A04J4195 to B.L.). X.X. acknowledges start-up grants from the Chinese Academy of Sciences.

## Author contributions

Q.Y. and X.X. conceived the study. Q.Y., R.H., and P.H. conceived and initiated the antibody isolation; X.X. conceived and initiated the structural studies; R.H. and P.H. isolated antibodies with assistance from X.H., H.L., X.C. and X.N.; P.H., X.H., H.Z., Y.Y. and R.H. expressed and purified antibodies; P.H., B.L., R.H. and Q.C. performed BLI assays; Y.M. performed the SARS-CoV-2 S-mediated cell-cell fusion assays; Ya. Z. performed authentic virus neutralization assays; B.L. and Q.C. purified spikes for cryo-EM and other experiments using constructs and protocols developed by X.X.; B.L. performed western blots to assay spike structural change; Q.Y. performed the bioinformatics analysis with assistance from X.X. and Yu. Z.; B.L. and X.G. collected cryo-EM data under the supervision of J.H. and X.X.; X.G., Z.L. and J.W. processed cryo-EM data under the supervision of J.H. and X.X.; X.X., X.G., Q.Y. and J.H. analyzed cryo-EM structures with assistance from B.L. and J.W.; Q.Y., B.L., X.G., R.H. and Y.M. prepared the figures under the supervision of X.X.; X.X. and Q.Y. wrote the paper with input from all co-authors; X.X., J.Z., J.H. and L.C. acquired funding and supervised the research.

## Competing interests

The authors declare no competing interests.

## Additional information

[1]State Key Laboratory of Respiratory Disease, National Clinical Research Center for Respiratory Disease, Guangzhou Institute of Respiratory Health, the First Affiliated Hospital of Guangzhou Medical University, Guangzhou, China. [2]Key Laboratory of Biological Targeting Diagnosis, Therapy and Rehabilitation of Guangdong Higher Education Institutes, The Fifth Affiliated Hospital of Guangzhou Medical University, Guangzhou, China. [3]State Key Laboratory of Respiratory Disease, Guangdong Provincial Key Laboratory of Stem Cell and Regenerative Medicine, Guangdong-Hong Kong Joint Laboratory for Stem Cell and Regenerative Medicine, Guangdong Provincial Key Laboratory of Biocomputing, Guangzhou Institutes of Biomedicine and Health, Chinese Academy of Sciences, Guangzhou, China. [4]Guangzhou Eighth People's Hospital, Guangzhou Medical University, Guangzhou, China. [5]Guangzhou National Laboratory, Guangzhou, China. [6]University of Chinese Academy of Sciences, Beijing, China. [7]Bioland Laboratory (Guangzhou Regenerative Medicine and Health - Guangdong Laboratory), Guangzhou, China. [8]Shanghai Institute for Advanced Immunochemical Studies, School of Life Science and Technology, ShanghaiTech University, Shanghai, China. [9]These authors contributed equally: Qihong Yan, Xijie Gao, Banghui Liu. ✉e-mail: he_jun@gibh.ac.cn; chen_ling@gibh.ac.cn; zhaojincun@gird.cn; xiong_xiaoli@gibh.ac.cn

