## [Peer Review File · Nature Communications]

Antibodies utilizing VL6-57 light chains target a convergent cryptic epitope on SARS-CoV-2 spike protein and potentially drive the genesis of Omicron variantsREVIEWER COMMENTS

Reviewer #1 (Remarks to the Author):

In this study, the authors describe a set of SARS-CoV-2 antibodies that use the VL6-57 germline gene, followed by characterization of the structure of one of those antibodies in complex with the spike, leading to comparative structural analysis that highlights a class of antibodies sharing that light chain gene and shared RBD targeting. The description of this class of antibodies and its features, in addition to its relevance to viral escape by Omicron, should be quite interesting to the community. Concerns regarding this manuscript are primarily regarding a need for more clarity or information in various elements of the results. Specific comments are noted below:

1. It seems that at least one previous study has identified a pre-pandemic VL6-57 antibody that neutralizes SARS-CoV-2: Bertoglio et al. Nature Communications 12:1577 2021 (Table 3). However, it does not appear to be cited or noted anywhere in the current study. The authors should cite that study, and discuss overlap or agreement between these studies.
2. In lines 110-113, the authors note obtaining 5 RBD-binding antibodies and cite the He et al. 2022 study. However, the He et al. study notes 6 mAbs in Extended Data Figure 1. To avoid confusion, the authors should note that 6 antibodies were identified in that study, and these antibodies represent the 5 mAbs that share that light chain gene.
3. In the Figure 1 legend, the authors state for panel (a): “The genetic and functional properties of the 5 isolated VL6-57 light chain utilizing antibodies (He et al., 2022)”. However, it is not clear to readers whether the antibodies themselves, or the data on the antibodies, are from the He et al. study. To clarify this point, the authors should state: “The genetic and functional properties of the 5 isolated VL6-57 light chain utilizing antibodies, as previously reported (He et al., 2022)”, because the data are from the He et al. study.
4. Line 236. The authors should state in the text the average heavy and light chain values (or percent average % BSA for the light chain, if they prefer) that they are referring to.
5. It seems that LCDR2 loop residues are involved in the RBD interaction for R1-26 (Fig. 2), but in Fig. 3 and in the Results text, it is not clear whether any LCDR2 loop RBD contacts are conserved among the VL6-57 antibody-RBD complex structures. As germline interactions were identified for the LCDR1 loop, the authors should state whether or not there were any

conserved interactions likewise identified for the LCDR2 loop, or if that loop is more variable in terms of its RBD interaction.

6. The LCDR1 loops do not seem clearly visible in Fig. 3 panels a-i. This may possibly be due to the loop shading and/or the limited resolution of the images, but the authors should double-check that those loops are clearly visible, and adjust the figures if needed, so that readers can see them as clearly as the HCDR3 loops.

7. Line 390. “recombinational” should probably be “recombinant”. This is true for later in the section (line 393) as well.

8. It seems that the authors describe individuals as “healthy”, “unexposed”, and “naïve” interchangeably. To avoid confusion and ambiguity, the authors should consistently use “naïve”, or “unexposed”, to describe those who have never been exposed or vaccinated.

9. Where the five unexposed repertoire antibodies are first described (lines 386-388) it should be clarified where they were obtained, i.e. by citing the specific previously published studies/datasets, if they were from one of the published datasets noted in the previous section (which seems to be the case).

10. In the title, the authors indicate that this class of antibodies is “driving the genesis of Omicron variants”, however that seems rather speculative as many other antibodies may be targeting that 371-373-375 RBD site within and between individuals. Thus, it is suggested that the authors consider changing this to “help to drive the genesis of Omicron variants”.

Reviewer #2 (Remarks to the Author):

In this study, Yan et al. identified a class of antibodies specific for SARS-CoV-2 Spike protein, characterized by VL6-57-encoded light chains, targeting a shared epitope defined by RBD residues S371-S373-S375 in the ancestral Spike protein. The aim of the study was to demonstrate that these antibodies may be associated with the immune pressure driving the emergence of S371L/F-S373P-S375F mutations in Omicron variants. Additionally, the article emphasized how the VL6-57 antibody-driven immune response was predominantly mediated by the light chain, and not as usual from the heavy chain, providing a unique example where a light chain can significantly influence the binding mode of a class of antiviral antibodies.

Overall, the study is well-organized and potentially interesting in the context of debates about the interplay between antibody immune pressure and the emergence of new SARS-CoV-2 mutations.

However, as a general comment, the article contains too many figures, and some of them appear redundant. Specifically, the results and discussion sections suffer from some important concerns and require significant strengthening.

Introduction:

1.Lines 99-100: The authors state “..the driving force behind the genesis of the S371L/F-S373P-S375F mutations remains enigmatic.”. Could the genesis of mutations altering virus tropism be driven by factors beyond immune pressure? The authors should better discuss this point, taking into consideration other aspects.

Results:

1.Lines 126-127: In the neutralization test, IC50 values for each antibody measured on WT spike based pseudovirions are mostly lower when compared to the Kd of the antibody affinity measured against RBD by BLI. While a correlation of the IC50 and Kd values on spike biological activity inhibition by antibodies or mini-binders is frequently observed, a lower IC50 value than Kd one is somewhat unusual, given the much more dynamic complexity of the on spike protein machinery. Would it be possible to ascribe these results only to the different techniques employed and experimental setups? In order to permit a comparison with the values reported in Figure 1a, would be possible to express the IC50 values panel 1e also in molar concentrations?

2.Line 185: The authors state “Notably, the canonical class 4 antibody CR3022 does not inhibit ACE2 binding and is only weakly neutralizing...” A class 4 antibody usually does not compete with RCM portion of RBD for ACE binding. However, since R1-26 unexpectedly exhibit neutralization properties, it is important to add appropriate references for this sentence.

3.Lines 190-194: Since the differential clash effect and entity by the glycan at position N322 of ACE-2 on CR3022 and R1-26 binding is mainly based on a modelling analysis inference, while reasonable, the concept should be attenuated. (See also comment to the relative Figure 2C)

4.Lines 200-204: The authors show that R1-26 possesses an activity to trigger fusogenic conformational change similar to class 1 mAbs including B38 and S230. However, in figures 2d and 2e, class 1 mAb S230 results are missing. These data should be shown in order to compare the various differences. Moreover, are there shared elements in the epitopes recognized by B38 and S230 that might indicate a mechanism for triggering a fusogenic conformational change? Is there a molecular mechanism that can explain the fusogenic activity?

5.Line 232: The authors analyzed 376 structurally characterized SARS-CoV-2-specific mAbs. Are all these structures experimentally determined or the structure of some of them has been predicted?

6.Line 279: It's really challenging to clearly identify N32 and Y33 residues in Figure 3 because of the image quality.

7.Lines 349-353: In figure 4F the authors observed clonal expansion of the heavy chains (WLRG) in COVID-19 patients by a longitudinal analysis post clinical onset. On the contrary, no clonal expansion was observed in the light chains of infected individuals. How do you explain this different evidence? Do you consider this result as expected? In my opinion, these are the only results within the study that support the hypothesis that antibodies utilizing VL6-57 light chains constitutes immune pressure promoting the introduction of S371L/F-S373P-S375F in Omicron variants. For this reason, the authors should deepen and broaden these findings.

8.Lines 384-402: Regarding to Figure S10, the authors should introduce other recombining experiments in the presence of other IGLs that are not VL6-57, as a negative control.

9.Line 418: The word “..disintegration..” might not be the correct term. The authors could likely use “spike trimer dissociation” or “spike protein disassembly”. Does it refer to the transition from the pre-fusion to the post-fusion state?

Discussion:

1.Line 489: Again, see comment number 9 in the results' section.

2.Lines 504-513: Does the lower pathogenicity of Omicron with a different tropism result from only these 3 mutations? Do common coronaviruses have a similar tropism and exhibit similar mutations? The authors should deeply discuss these observations.

3.In the discussion at lines 571-573 the authors state “Our findings further support the

hypothesis that convergent antibody responses within the population drive viral antigenic drift leading to emergence of new SARS-CoV-2 variants.”, as well as in the abstract at lines 43-45 “These findings support that this class of public antibodies constitutes immune pressure promoting the introduction of S371L/F-S373P-S375F in Omicron variants.” and in the title “Antibodies utilizing VL6-57 light chains target a convergent cryptic epitope on SARS-CoV-2 spike protein driving the genesis of Omicron variants”. In my opinion, these statements are mainly and exclusively reflected in the experiments conducted in the infected population (Figure 4, see comment number 7 in the results’ section). For this reason, I believe that they should reconsider these conclusions.

4. What is the potential application of this discovery in the future development of new generation, more potent COVID-19 vaccines?

Figure:

1. Figure 2C: The green dashed arrow, indicating a possible direction of glycan chain extension, should be represented in both structures.
2. Figure 2E: Are the comparisons not significant? class 1 mAb S230 results are missing.
3. Figure S5: 36 structures sharing the characteristic of belonging to class 4 antibodies are shown in Figure S5, however it’s hard to analyze them due to the complexity and the huge number of the different structures. The authors should better highlight at least the 11 structures of antibodies encoded by IGLV6-57.

Reference:

1. Line 960: This is an outdated reference for Dong and associates. The latest manuscript was published in Nature Microbiology, September 2021.

Minor points

Introduction:

1. Lines 69-72: In this sentence "recurrent mutations" is repeated.
2. Lines 73-74: It would be better to replace the sentence “..the epitopes of several classes of germline antibodies..” with the new one “..the epitopes recognized by several classes of germline antibodies..”

3.Lines 313-318: This sentence is too long and and not easily understood. Please, simplify or break up the sentence.

Reviewer #1 (Remarks to the Author):

In this study, the authors describe a set of SARS-CoV-2 antibodies that use the VL6-57 germline gene, followed by characterization of the structure of one of those antibodies in complex with the spike, leading to comparative structural analysis that highlights a class of antibodies sharing that light chain gene and shared RBD targeting. The description of this class of antibodies and its features, in addition to its relevance to viral escape by Omicron, should be quite interesting to the community. Concerns regarding this manuscript are primarily regarding a need for more clarity or information in various elements of the results. Specific comments are noted below:

Response: We thank the reviewer for the above positive comments and the constructive feedbacks. We have carefully modified the manuscript according to the comments to address the reviewer's concerns.

1. It seems that at least one previous study has identified a pre-pandemic VL6-57 antibody that neutralizes SARS-CoV-2: Bertoglio et al. Nature Communications 12:1577 2021 (Table 3). However, it does not appear to be cited or noted anywhere in the current study. The authors should cite that study, and discuss overlap or agreement between these studies.

Response: We thank the reviewer for bringing this publication into our attention. We apologize for the omission of this reference. Inspired by the reviewer's comment, we have done an in-depth literature search and we found that 3 publications have mentioned about SARS-CoV-2 S-specific VL6-57 antibodies in the context of their wide presence:

(1) Robbiani, D. F., et al. (2020). "Convergent antibody responses to SARS-CoV-2 in convalescent individuals." Nature 584(7821): 437-442.

(PMID: 32555388)

(2) Bertoglio, F., et al. (2021). "SARS-CoV-2 neutralizing human recombinant antibodies selected from pre-pandemic healthy donors binding at RBD-ACE2 interface." *Nature Communications* 12(1). (PMID: 33707427)

The two papers above simply observed that SARS-CoV-2 S-specific antibodies, isolated using single B-cell PCR and phage display techniques, respectively, show an increased representation of antibodies utilizing the light chain gene VL6-57. The two studies did not investigate the molecular mechanism behind such phenomena.

(3) Wang, Y., et al. (2022). "A large-scale systematic survey reveals recurring molecular features of public antibody responses to SARS-CoV-2." *Immunity* 55(6): 1105-1117.e1104. (PMID: 35397794)

This study comprehensively surveyed isolated SARS-CoV-2 S-specific antibodies. It also noted that VL6-57 antibodies are over-represented in the isolated antibodies, noting that the HCDR3s of VL6-57 antibodies tend to have a "WLRG" motif. The study hypothesized that VL6-57 antibodies contribute to RBD-specific public antibody response without diving into the mechanism.

We now cite the three studies in our result section “VL6-57 antibodies paired with HCDR3 “WLRG” motif are widely induced in the population and clonally expanded upon SARS-CoV-2 infection”, with the following comments (In bold) at Line 386-395 and Line 417-423 in the track-change manuscript:

“HCDR3 sequence analysis shows that there is a strong preference for a 12-AA long HCDR3 containing the “WLRG” motif (68/99 mAbs) as observed in S2A4, P5S-3B11, 3D11, R1-30, R2-3, R2-6 and R2-7 (Figs. 4b, S6b). **A previous survey also noticed that the isolated SARS-CoV-2 S-specific VL6-57 antibodies tend to have a “WLRG” motif at the tip of the HCDR3 (Wang et. al. 2022b)**. In addition, we also found a small percentage of mAbs (7/99) containing a 12-AA long HCDR3 with the “YYY” motif observed in 553-15 (Figs. 4b logo plot, S6b).” (Line 386-395)

and

“The above analysis suggests that the “WLRG” motif within the 12-AA long HCDR3 and the “QSYDSS” motif within the LCDR3 are the convergent signatures of the SARS-CoV-2 RBD-specific VL6-57 mAbs, which may be widely induced in the COVID-19 population. **Such proposal may explain previous observations that antibodies utilizing the light chain gene IGLV6-57 are over-represented in the isolated SARS-CoV-2 S-specific antibodies (Robbiani et al., 2020; Bertoglio et al., 2021; Wang et. al. 2022).**” (Line 417-423)

We also commented in the Discussion section:

“This study provides a rare example that a light chain paired with common binding motifs in various heavy chains can largely dictate the binding mode of a class of antiviral antibodies and **provides a molecular mechanism to explain the increased identification of VL6-57 S-specific mAbs among SARS-CoV-2 infected individuals**

(Bertoglio et al., 2021; Robbiani et al., 2020; Wang et al., 2022b).” (Line 767-772)

2. In lines 110-113, the authors note obtaining 5 RBD-binding antibodies and cite the He et al. 2022 study. However, the He et al. study notes 6 mAbs in Extended Data Figure 1. To avoid confusion, the authors should note that 6 antibodies were identified in that study, and these antibodies represent the 5 mAbs that share that light chain gene.

Response: We thank the reviewer for pointing out this potential source of confusion. We have amended our manuscript for better accuracy and clarity as follows:

“we have reported the isolation of 6 mAbs, namely R1-26, R1-30, R1-32, R2-3, R2-6, and R2-7 by phage display using SARS-CoV-2 RBD as the bait (He et al., 2022). Interestingly, five out of the six isolated antibodies utilize light chains encoded by IGLV6-57, while their heavy chains are of different genetic origins (Fig. 1a)” (Line 120-158).

3. In the Figure 1 legend, the authors state for panel (a): “The genetic and functional properties of the 5 isolated VL6-57 light chain utilizing antibodies (He et al., 2022)”. However, it is not clear to readers whether the antibodies themselves, or the data on the antibodies, are from the He et al. study. To clarify this point, the authors should state: “The genetic and functional properties of the 5 isolated VL6-57 light chain utilizing antibodies, as previously reported (He et al., 2022)”, because the data are from the He et al. study.

Response: We thank the reviewer for this suggestion. We have revised the Figure 1 legend as suggested. The revised legend reads:

“The genetic and functional properties of the 5 isolated VL6-57 light chain utilizing antibodies, as previously reported (He et al., 2022)” (Line 1541-1542).

4. Line 236. The authors should state in the text the average heavy

and light chain values (or percent average % BSA for the light chain, if they prefer) that they are referring to.

Response: We thank the reviewer for this suggestion. We have included the average BSA values of the heavy and light chains in the revised manuscript.

The revised statement reads:

“heavy chains dominate epitope interaction, burying significantly more surface areas than light chains (HC vs. LC: 613 ± 173 Å vs. 283 ± 149 Å) (Fig. S7a). Different from typical SARS-CoV-2 S-specific mAbs, the average BSA of VL6-57 class 4 mAbs is comparable between heavy and light chains (HC vs. LC: 466 ± 165 Å vs. 418 ± 104 Å) (Fig. S7b).” (Line 311-314)

5. It seems that LCDR2 loop residues are involved in the RBD interaction for R1-26 (Fig. 2), but in Fig. 3 and in the Results text, it is not clear whether any LCDR2 loop RBD contacts are conserved among the VL6-57 antibody-RBD complex structures. As germline interactions were identified for the LCDR1 loop, the authors should state whether or not there were any conserved interactions likewise identified for the LCDR2 loop, or if that loop is more variable in terms of its RBD interaction.

Response: We thank the reviewer for this insightful comment. Indeed, R1-26 LCDR2 residues E51 and D52 have been identified to form interactions with RBD residues K378 and T385 respectively. Although LCDR2 are conserved among the R1-26-like VL6-57 antibodies, our original interaction analysis (old Fig. S7c) found that the LCDR2 of antibody 2-7 was unable to form interaction with the RBD. Therefore, we arrived at the conclusion that interactions between the LCDR2s and RBDs vary among different VL6-57 antibodies, and we refrained from commenting on LCDR2 interactions for antibodies other than R1-26. Prompt by the

comment, we analyzed the antibody complex structures again and we found that there are inaccuracies in the structural modelling of the LCDR2 of antibody 2-7 and 553-15. We remodeled the LCDR2 of antibody 2-7 (PDB:7T3M) and 553-15 (PDB: 7WO7), based on the deposited corresponding experimental cryo-EM maps (EMDB: EMD-25663 (2-7) and EMD-32641 (553-15)). We now find that all R1-26-like VL6-57 antibodies have highly similar LCDR2 interactions to RBD. LCDR2 residues E51 and D52 form conserved interactions to RBD residues T385/S383 and K378 respectively.

We now comment:

“For LCDR2, the germline-encoded E51 and D52 shared among the 8 R1-26-like mAbs form conserved interactions to RBD residues T385/S383 and K378 respectively, as observed in R1-26 (Fig. 2b,5f, S6c, S7c).” (Line 367-370)

6. The LCDR1 loops do not seem clearly visible in Fig. 3 panels a-i. This may possibly be due to the loop shading and/or the limited resolution of the images, but the authors should double-check that those loops are clearly visible, and adjust the figures if needed, so that readers can see them as clearly as the HCDR3 loops.

Response: We thank the reviewer for pointing out this issue. We have changed the lighting settings in the molecular graphics program Chimera to dampen the shadows on LCDR1 loops and we have moved the label text boxes away from the LCDR1 loops so that the LCDR1 loops should have much better visibility in the revised Fig.3 a-i (Line 1594).

7. Line 390. "recombinational" should probably be "recombinant". This is true for later in the section (line 393) as well.

Response: We thank the reviewer for this suggestion. We have revised the manuscript texts at the two places accordingly.

8. It seems that the authors describe individuals as "healthy", "unexposed", and "naïve" interchangeably. To avoid confusion and ambiguity, the authors should consistently use "naïve", or "unexposed", to describe those who have never been exposed or vaccinated.

Response: We thank the reviewer for pointing out this potential source of confusion. In the revised manuscript, we use "naïve" consistently to describe those who have never been exposed or vaccinated.

9. Where the five unexposed repertoire antibodies are first described (lines 386-388) it should be clarified where they were obtained, i.e. by citing the specific previously published studies/datasets, if they were from one of the published datasets noted in the previous section (which seems to be the case).

Response: We thank the reviewer for this comment. These antibodies were cloned from bulk VDJ-sequencing datasets previously published by our research team. We have revised the text as:

“five different “WLRG” motif-containing heavy chains encoded by IGHV3-7, IGHV3-11, IGHV4-31, IGHV4-39 and IGHV4-61 respectively from 5 different naïve donors according to previously published VDJ-sequencing datasets by our research team (Niu et al., 2020; Yan et al., 2021; Zhang et al., 2022)”

to clearly indicate the source of the sequences (Line 494-497).

10. In the title, the authors indicate that this class of antibodies is "driving the genesis of Omicron variants", however that seems

rather speculative as many other antibodies may be targeting that 371-373-375 RBD site within and between individuals. Thus, it is suggested that the authors consider changing this to "help to drive the genesis of Omicron variants".

Response: We thank the reviewer for this suggestion. We have modified the title according to the suggestion as:

“Antibodies utilizing VL6-57 light chains target a convergent cryptic epitope on SARS-CoV-2 spike protein and potentially drive the genesis of Omicron variants” (Line 1-2)

Reviewer #2 (Remarks to the Author):

In this study, Yan et al. identified a class of antibodies specific for SARS-CoV-2 Spike protein, characterized by VL6-57-encoded light chains, targeting a shared epitope defined by RBD residues S371-S373-S375 in the ancestral Spike protein. The aim of the study was to demonstrate that these antibodies may be associated with the immune pressure driving the emergence of S371L/F-S373P-S375F mutations in Omicron variants. Additionally, the article emphasized how the VL6-57 antibody-driven immune response was predominantly mediated by the light chain, and not as usual from the heavy chain, providing a unique example where a light chain can significantly influence the binding mode of a class of antiviral antibodies. Overall, the study is well-organized and potentially interesting in the context of debates about the interplay between antibody immune pressure and the emergence of new SARS-CoV-2 mutations. However, as a general comment, the article contains too many figures, and some of them appear redundant. Specifically, the results and discussion sections suffer from some important concerns and require significant strengthening.

Response: We thank the reviewer for the overall positive comment on the manuscript and his/her thoughtful and attention-to-detail comments. We have carefully considered the reviewer’s concerns and specific comments, we made the following modifications to improve the manuscript.

Introduction:

1. Lines 99-100: The authors state "...the driving force behind the genesis of the S371L/F-S373P-S375F mutations remains enigmatic.". Could the genesis of mutations altering virus tropism be driven by factors beyond immune pressure? The authors should better discuss this point, taking into consideration other aspects.

Response: We thank the reviewer for this suggestion. Influenced by earlier reports, we originally perceived that S371L/F-S373P-S375F are the primary change led to the change of tissue tropism in Omicron variants. After careful study of most recent publications, we conclude that altered virus phenotypic changes (inducing tissue tropism) could result from multiple factors, not just immune pressure, potentially leading to enhanced virus transmissibility in the population.

In the revised discussion section, we first summarize known functions of S371L/F-S373P-S375F, including their abilities to affect spike processing and pseudovirus entry (Kimura et al., 2022; PMID: 36507224. Pastorio et al., 2022; PMID: 35931073). A recent published study has demonstrated that S375F is able to promote immune evasion by enhancing spike binding to Siglec-9 on macrophages to impair antigen phagocytosis and antigen presentation (He et al., 2024; PMID: 36151403). A recent preprint also shows that G339H, S371F, S373P and S375F are also important for authentic virus epithelium cell tropism (Furnon et al., 2024; doi: 10.21203/rs.3.rs-4283987/v1). Multiple reports also suggest a complex epistatic relationship between S-protein mutations, therefore, S371L/F,

S373P and S375F may only be functional when they co-occur with the other Omicron specific S-protein substitutions (Furnon et al., 2024; doi: 10.21203/rs.3.rs-4283987/v1. Martin et al., 2022; PMID: 35325204). Literature search also found that other factors affect tissue tropism of SARS-CoV-2, several reports describe the use of membrane-type matrix metalloproteinases (MT-MMPs) and ADAM (a disintegrin and metalloproteinase) being important for nasal cell tropism (Chan et al., 2023; PMID: 36662861. Jocher et al., 2022; PMID: 35527514. Shi et al., 2024; PMID: 38291024). Tissue tropism and pathogenesis determinants have also been attributed to host interferon responses (Bouhaddou et al., 2023; PMID: 37738970. Kehrer et al., 2023; 37738983. Lista et al., 2022; PMID: 36350154. Mesner et al., 2023; PMID: 36693093; Reuschl et al., 2024; PMID: 38228858. Shi et al., 2024; PMID: 38291024. Thorne et al., 2022; PMID: 34942634). Some Omicron attenuation determinants have been specifically mapped outside of the S-protein gene (Chen et al., 2023; PMID: 36630998. Liu et al., 2022; PMID: 36075211).

Based on the above literature search, we conclude that the correlation between SARS-CoV-2 tissue tropism, pathogenicity, immune evasion, and transmissibility is highly complex and remains ill-defined.

The first two paragraphs of the Discussion section, in particular the section paragraph (shown below) have been revised to reflect the above

conclusions.

“The above observations highlight the likely necessity of the S371-S373-S375 mutations for SARS-CoV-2 Omicron variants, even at the expense of drastic S-protein phenotypic change with associated impact on virus behaviour. It has been shown that antigenic changes can alter SARS-CoV-2 receptor binding (Niu et al., 2021), for other viruses, antigenic evolution of the pandemic H3N2 virus was found to substantially weaken hemagglutinin (HA) sialic acid binding affecting virological phenotypes (Lin et al., 2012). While our study provides evidence to show that the S371-S373-S375 mutations enable SARS-CoV-2 to evade the identified class of VL6-57 public antibodies, other functions for these mutations continue to emerge. A recent study reports that S375F promotes immune evasion by enhancing spike binding to Siglec-9 on macrophages to impair antigen phagocytosis and presentation (He et al., 2024). A recent preprint also shows that when G339H, S371F, S373P and S375F mutations are reverted in a reverse-genetic authentic virus carrying Omicron BA.1 S-protein in an ancestral virus genetic background, the Omicron nasal epithelium cell tropism is almost completely abrogated (Furnon et al., 2024). However, a complex epistatic relationship between S-protein mutations has been suggested – S371L/F, S373P and S375F may only be adaptive when they co-occur with the other Omicron specific S-protein substitutions (Furnon et al., 2024; Martin et al., 2022). Indeed, the correlation between SARS-CoV-2 tissue tropism, pathogenicity, immune evasion, and transmissibility is highly complex and remains ill-defined. Other factors affecting SARS-CoV-2 infection continue to emerge, utilization of membrane-type matrix metalloproteinases (MT-MMPs) and ADAM (a disintegrin and metalloproteinase) metalloproteinases has recently been proposed to correlate with nasal cell tropism (Chan et al., 2023; Jocher et al., 2022; Shi et al., 2024). Other Omicron tissue tropism and pathogenesis determinants related to host interferon responses have been recently implicated (Bouhaddou et al., 2023; Kehrer et al., 2023; Lista et al., 2022; Mesner et al., 2023; Reuschl et al., 2024; Shi et al., 2024; Thorne et al., 2022). Some Omicron attenuation determinants have been mapped outside of the S-protein gene (Chen et al., 2023; Liu et al., 2022b). Nevertheless, multiple functions identified so far for the S371L/F, S373P and S375F mutations illustrate vividly that antigenic change of viral surface protein can have complex consequences and a delicate balance among tissue tropism, pathogenicity, immune evasion, and transmissibility is likely pursued by circulating viruses (Carabelli et al., 2023).” (Line 654-703)

Results:

1. Lines 126-127: In the neutralization test, IC50 values for each antibody measured on WT spike based pseudovirions are mostly lower when compared to the Kd of the antibody affinity measured against RBD by BLI. While a correlation of the IC50 and Kd values on spike biological activity inhibition by antibodies or mini-binders is

frequently observed, a lower IC₅₀ value than K_d one is somewhat unusual, given the much more dynamic complexity of the on spike protein machinery. Would it be possible to ascribe these results only to the different techniques employed and experimental setups? In order to permit a comparison with the values reported in Figure 1a, would be possible to express the IC₅₀ values panel 1e also in molar concentrations?

Response: We thank the reviewer for this comment. We completely agree with the reviewer that it would be very unusual for an IC₅₀ values to be lower than the corresponding K_D! We apologize for the confusion.

It should be noted that the binding affinities (K_{DS}) reported in Fig. 1a were determined through biolayer interferometry (BLI) by immobilizing antibodies in the IgG form onto the protein A BLI biosensors and dipping the biosensors into analyte solutions containing different concentrations of monomeric RBDs (original data is in Extended Data Fig. 1 of He et. al., PMID: 36151403). In this set-up, because of the RBDs being monomeric, IgG binding sites are independent and the measured binding affinities (K_{DS}) are for individual sites without any amplification by the avidity effect. The IC₅₀ values reported in Fig. 1a were determined in neutralization assays using pseudoviruses displaying spike trimers (S-trimers). In this setup, the antibody used is dimeric IgG and the antigen is S-trimer. Therefore, the trimeric antigen has the ability to cross-link the two antigen binding sites of an IgG, as a result, antibody binding in virus neutralization assays is subject to amplification by the avidity effect. The

avidity effect enhanced antibody binding can be estimated by BLI assays using analyte solutions containing S-trimers instead of RBDs.

Unfortunately, we don't have the avidity enhanced apparent K_{DS} determined for R1-30, R2-3, R-26 and R2-7 IgGs towards the wildtype S-trimer. For R1-26, we do have the K_{DS} determined for the WT, Alpha, Beta, Delta and Omicron BA.1 RBDs (Fig. 1c) and the corresponding S-trimers (Fig. 1d). The determined K_{DS} of R1-26 IgG are 3.84, 11.05, 9.78, 10.29 nM, respectively, towards the WT, Alpha, Beta, Delta RBDs, and non-detectable for the Omicron BA.1 RBD. The avidity enhanced R1-26 binding towards S-trimers is very slowly dissociating and the apparent K_{DS} are at least 0.36, 3.04, 1.22, 0.17 and 48 nM (revised Fig. 1) and can be estimated to the upper bounds of 0.058, 0.021, 0.0083, 0.0059 and 16 nM (in Supplementary Table 1), respectively, towards WT, Alpha, Beta, Delta and Omicron BA.1 S-trimers. Therefore, the avidity effect can enhance antibody binding by at least 3- to 10-fold, but likely much more. Therefore, the true avidity enhanced affinities between dimeric IgGs and S-trimers, as would apply in a neutralization assay, would be substantially tighter, than those affinities estimated using monomeric RBDs as reported in Fig.1a. However, we still prefer to report the K_{DS} for RBDs for ease of comparison across different antibodies because, as illustrated by the R1-26 S-trimer binding data, the avidity effect often results in varying

degrees of affinity amplification due to factors that are very difficult to characterize.

We have modified Fig. 1d to show avidity enhanced K_D lower bounds for S-trimer binding data. This highlights the enhanced avidity of IgG binding to S-trimers compared to monomeric RBDs.

2. Line 185: The authors state "Notably, the canonical class 4 antibody CR3022 does not inhibit ACE2 binding and is only weakly neutralizing..." A class 4 antibody usually does not compete with RCM portion of RBD for ACE binding. However, since R1-26 unexpectedly exhibit neutralization properties, it is important to add appropriate references for this sentence.

Response: We thank the reviewer for this suggestion. we have revised the manuscript to include relevant reference to support the statement about the neutralization properties of CR3022 and R1-26. The revised statement reads:

"Notably, the canonical class 4 antibody CR3022 does not inhibit ACE2 binding and is only weakly neutralizing (He et al., 2022; Jette et al., 2021). In contrast, strong ACE2 binding inhibition is observed for R1-26 (Fig. 1b) and other class 4 antibodies such as C118 and C022 (Jette et al., 2021), as well as S2A4 and S2X35 (Piccoli et al., 2020)" (Line 233-237)

(1) Jette, C. A., et al. (2021). "Broad cross-reactivity across sarbecoviruses exhibited by a subset of COVID-19 donor-derived neutralizing antibodies." Cell Reports 36(13).

(2) He, P., et al. (2022). "SARS-CoV-2 Delta and Omicron variants evade

population antibody response by mutations in a single spike epitope." *Nat Microbiol* 7(10): 1635-1649.

(3) Piccoli, L., et al. (2020). "Mapping Neutralizing and Immunodominant Sites on the SARS-CoV-2 Spike Receptor-Binding Domain by Structure-Guided High-Resolution Serology." *Cell* 183(4): 1024-1042 e1021.

3.Lines 190-194: Since the differential clash effect and entity by the glycan at position N322 of ACE-2 on CR3022 and R1-26 binding is mainly based on a modelling analysis inference, while reasonable, the concept should be attenuated. (See also comment to the relative Figure 2C)

Response: We thank the reviewer for this suggestion. We attenuated the modelling result by moving the results into the Supplementary Material section Fig. S4. While we feel that our description of the modelling result is objective and we keep it as it is, we have added references to 2 previous studies which also linked neutralization activities of Class 4 antibodies to direct ACE2 binding competition, which is in line with the concept we propose here. The section now reads below:

“Notably, the canonical class 4 antibody CR3022 does not inhibit ACE2 binding and is only weakly neutralizing (Jette et al., 2021). In contrast, strong ACE2 binding inhibition is observed for R1-26 (Fig. 1b) and other class 4 antibodies such as C118 and C022 (Jette et al., 2021), as well as S2A4 and S2X35 (Piccoli et al., 2020). By modelling, we found that the approach angle of R1-26 is more tilted towards the modelled ACE2 bound to RBD (Fig. S4a), while the approach angle of CR3022 is more tilted away from it (Fig. S4b). In addition, R1-26 and CR3022 adopt different orientations when bound to RBD (Fig. S4a-b). Due to the differences in RBD binding, simultaneous binding of R1-26 and ACE2 to RBD would result in steric clashes between R1-26 and the glycan chain attached to ACE2 residue N322 (Fig. S4a). By contrast, simultaneous binding of CR3022 and ACE2 is possible likely due to much weaker clash (Fig. S4b), consistent with previous results of competition assays (Tian et al., 2020). Differences in RBD binding between R1-

26 and CR3022 likely confer superior ACE2 blocking activity to R1-26. **A few studies have also linked the neutralization activities of class 4 antibodies (including VL6-57 antibodies S2A4) to their ability to block ACE2 binding (Jette et al., 2021; Piccoli et al., 2021).**” (Line 233-250)

We hope this should provide more support to the concept we proposed.

4. Lines 200-204: The authors show that R1-26 possesses an activity to trigger fusogenic conformational change similar to class 1 mAbs including B38 and S230. However, in figures 2d and 2e, class 1 mAb S230 results are missing. These data should be shown in order to compare the various differences. Moreover, are there shared elements in the epitopes recognized by B38 and S230 that might indicate a mechanism for triggering a fusogenic conformational change? Is there a molecular mechanism that can explain the fusogenic activity?

Response: We thank the reviewer’s comment and we apologize for any confusion caused. B38 is a SARS-CoV-2 S-specific antibody that recognizes a Class 1 epitope, whereas S230 is a SARS-CoV-1 S-specific antibody, with an epitope that can also be classified as Class 1, if the SARS-CoV-2 epitope classification convention is followed. Class 1 epitope is defined as an epitope that highly overlaps the binding footprint of ACE2 and directly competing with ACE2 binding (Barnes et al., 2020 PMID: 33045718). In a previous study, we found that S230, with a binding footprint highly similar to that of ACE2, is able to trigger a fusogenic conformational change of the SARS-CoV-1 S-trimer (Walls, A.C et al., 2019; PMID: 30712865).

Regarding the shared epitopes recognized by B38 and S230, previous

structural and epitope mapping studies suggest that both antibodies target regions highly overlapping the ACE2 binding sites on the SARS-CoV-2 and SARS-CoV-1 S-proteins, respectively. Although the exact RBD residues involved differ and the two antibodies are unable to cross-react between SARS-CoV-2 and SARS-CoV-1 S-proteins. Nevertheless, we previously ascribed S230's ability to trigger the S-protein fusogenic conformational change to its ability to promote the S-trimer RBDs to adopt an "up" or "open" conformation. Based on the observation that ACE2 binding is also able to promote S-trimer RBDs to adopt an "up" conformation and that S230 and ACE2 share very similar binding footprints on S-RBD, we concluded that an antibody with an epitope similar to the footprint of ACE2 on S-RBD can mimic the ACE2 receptor to promote RBDs to open, triggering the fusogenic conformational change (Walls, A.C et al., 2019; PMID: 30712865). Since the pandemic, with many more S-specific antibodies being isolated for SARS-CoV-2, we confirmed in He et. al. (PMID: 36151403) that Class 1 SARS-CoV-2 S-specific antibodies (B38 and rmAb23) are indeed able to trigger the fusogenic conformational change of the SARS-CoV-2 S-protein, similar to S230 and its effect on the SARS-CoV-1 S-protein. In our previous study (He et. al. (PMID: 36151403)), we further found that R1-32, a non-ACE2 competing antibody, is also able to promote spike RBDs to open but is unable to trigger a fusogenic conformational change; instead, it causes S-trimer

disassembly. Therefore, we speculate that the specific geometry of the S-trimer's transition into an RBD "up", open conformation, may be important for the S-trimer to enter into a fusogenic conformational change. It is very interesting that, in this case, both Class 4 antibodies R1-26 and CR3022, which bind to similar epitopes, behave differently: R1-26 competes with ACE2 and can trigger a fusogenic conformational change, while CR3022, which does not compete with ACE2, cannot. The exact reason for the above different behaviours likely requires further more in-depth investigation.

The following publications are cited to clarify the above comment.

(1) Walls, A.C et al., (2019). Unexpected Receptor Functional Mimicry Elucidates Activation of Coronavirus Fusion. *Cell* 176, 1026-1039 e1015.

(2) Barnes, C. O., et al. (2020). "SARS-CoV-2 neutralizing antibody structures inform therapeutic strategies." *Nature* 588(7839): 682-687.

(3) He, P., et al. (2022). "SARS-CoV-2 Delta and Omicron variants evade population antibody response by mutations in a single spike epitope." *Nat Microbiol* 7(10): 1635-1649.

(4) Ge J, et al., (2021). Antibody neutralization of SARS-CoV-2 through ACE2 receptor mimicry. *Nat Commun.* Jan 11;12(1):250.

5. Line 232: The authors analyzed 376 structurally characterized SARS-CoV-2-specific mAbs. Are all these structures experimentally determined or the structure of some of them has been predicted?

Response: We would like to clarify that all the structures analyzed in our

study are experimentally determined, which are deposited in and publicly available from the Protein Data Bank (PDB). There are no predicted structures included in our analysis. We have added a statement to clarify that:

“To further understand antigen binding by VL6-57 light chain utilizing mAbs, we surveyed the Protein Data Bank (PDB) for structurally characterized VL6-57 light chain utilizing mAbs bound to SARS-CoV-2 S-protein or RBD. Among a dataset of 376 SARS-CoV-2 S-specific mAb complex structures, **which are all experimentally determined by X-ray crystallography or cryo-EM**, 12 mAbs utilize VL6-57 encoded light chains” (Line 289-294).

6. Line 279: It's really challenging to clearly identify N32 and Y33 residues in Figure 3 because of the image quality.

Response: We apologize for the difficulty in reading the original Fig. 3. We have adjusted the lighting settings in the molecular graphics program Chimera to dampen the shadows on LCDR1 loops and we have moved the label text boxes away from the LCDR1 loops so that N32 and Y33 in LCDR1 loops should have much better visibility in the modified Fig. 3 to address this issue.

7. Lines 349-353: In figure 4F the authors observed clonal expansion of the heavy chains (WLRG) in COVID-19 patients by a longitudinal analysis post clinical onset. On the contrary, no clonal expansion was observed in the light chains of infected individuals. How do you explain this different evidence? Do you consider this result as expected? In my opinion, these are the only results within the study that support the hypothesis that antibodies utilizing VL6-57 light chains constitutes immune pressure promoting the introduction of S371L/F-S373P-S375F in Omicron variants. For this reason, the

authors should deepen and broaden these findings.

Response: We thank the reviewer for this insightful comment. There may be two possible reasons for this observation: one possibility is that the high prevalence of VL6-57 in the total bulk antibody repertoire of uninfected individuals suggests that other antibodies also utilize the VL6-57 light chain. After SARS-CoV-2 infection, although there is a certain clonal expansion of B cells expressing WLRG/VL6-57, this expansion may not be as evident in the total VL6-57 repertoire with a high baseline. Another possibility is that the sequencing data for the light chain mainly came from samples taken within 2 weeks after infection (at later time points, only heavy chain sequencing was performed, and antibody light chain library sequencing was not conducted because of the limited amount of PBMCs), and the clonal expansion within 2 weeks may not be as pronounced. The WLRG heavy chain sequence shows significant clonal expansion between 2-4 weeks, with less pronounced clonal expansion within the first 2 weeks after infection (Fig. 4f).

To ensure the reliability of our conclusions, we conducted an analysis as presented in Fig. 4J using a collection of previously published naïve or SARS-CoV-2 exposed heavy-light paired V(D)J datasets from different labs. The result confirms that there was a significant clonal expansion of B cells expressing WLRG/VL6-57 after SARS-CoV-2 infection. Although our data have certain limitations, several previous studies analyzing the

frequency of light chain usage in S-specific antibodies have also noticed a significant increase in VL6-57 usage (Bertoglio et al., 2021; Robbiani et al., 2020; Wang et al., 2022). These reports also support our conclusions, albeit indirectly.

(1) Bertoglio, F., et al. (2021). "SARS-CoV-2 neutralizing human recombinant antibodies selected from pre-pandemic healthy donors binding at RBD-ACE2 interface." *Nature Communications* 12(1).

(2) Robbiani, D. F., et al. (2020). "Convergent antibody responses to SARS-CoV-2 in convalescent individuals." *Nature* 584(7821): 437-442.

(3) Wang, Y., et al. (2022). "A large-scale systematic survey reveals recurring molecular features of public antibody responses to SARS-CoV-2." *Immunity* 55(6): 1105-1117.e1104.

8.Lines 384-402: Regarding to Figure S10, the authors should introduce other recombining experiments in the presence of other IGLs that are not VL6-57, as a negative control.

Response: We thank the reviewer for this valuable suggestion. In our revised manuscript, we have incorporated new experimental data with non-VL6-57 IGLs from S309 and CR3022 to serve as negative controls (9 out of 10 tested negative control mAbs were successfully expressed). The new recombination experiments are shown in Fig. S10d-e. Based on the result, a sentence was added:

“Non-VL6-57 light chains of S309 (VK3-20) and CR3022 (VK4-1) were also used to pair with the selected heavy chains to generate recombinant mAbs as negative controls. Apart

from the pairing of H18-H & CR3033-L, the other 9 recombinant mAbs were successfully expressed (Fig. S10d). As expected, ELISA assays show that these 9 mAbs are unable to bind Spike RBD (Fig. S10e).” (Line 504-509)

We believe that these negative controls will strengthen the robustness of our findings.

9. Line 418: The word “..disintegration..” might not be the correct term. The authors could likely use “spike trimer dissociation” or “spike protein disassembly”. Does it refer to the transition from the pre-fusion to the post-fusion state?

Response: We appreciate the reviewer’s suggestion. We have changed the term “disintegration” to “S-trimer disassembly”. Although spike inactivation can result from both premature triggering of the spike (premature spike transition from the pre-fusion to the post-fusion state) and S-trimer disassembly. “S-trimer disassembly” is not the same as the transition from the pre-fusion to the post-fusion state. It’s known that the postfusion spike has a defined structure while disassembled S-trimer is largely disordered. Where applicable we discuss the two processes separately, for example, in line 620-626:

“Indeed, we find that binding of R1-26, H18 and possibly other VL6-57 mAbs to the convergent cryptic epitope not only blocks ACE2 binding (Piccoli et al., 2020), but also prematurely triggers spike conformational change resulting in spike inactivation. When paired with other RBD-specific antibodies, binding of antibodies to cryptic epitopes may also result in spike inactivation by promoting S-trimer disassembly (He et al., 2022; Yu et al., 2023)” (Line 620-626)

Discussion:

1.Line 489: Again, see comment number 9 in the results’ section.

Response: We have modified the term "disintegration" to "S-trimer disassembly".

2. Lines 504-513: Does the lower pathogenicity of Omicron with a different tropism result from only these 3 mutations? Do common coronaviruses have a similar tropism and exhibit similar mutations? The authors should deeply discuss these observations.

Response: We thank the reviewer for this comment. Tissue tropism of common cold coronaviruses remains under-studied, a recent report:

Otter, C.J., Fausto, A., Tan, L.H., Khosla, A.S., Cohen, N.A., Weiss, S.R. (2023) Infection of primary nasal epithelial cells differentiates among lethal and seasonal human coronaviruses. *Proc Natl Acad Sci U S A* 120: e2218083120

suggests that virus infection characteristics in nasal epithelial cells influence downstream infection outcomes such as disease severity and transmissibility for non-SARS-CoV-2 human coronaviruses, in line with observations on SARS-CoV-2.

Based on several earlier reports, examples were listed below:

(1) Suzuki, R., et al. (2022). "Attenuated fusogenicity and pathogenicity of SARS-CoV-2 Omicron variant." *Nature* 603(7902): 700-705.

(2) Meng, B., et al. (2022). "Altered TMPRSS2 usage by SARS-CoV-2 Omicron impacts infectivity and fusogenicity." *Nature* 603(7902): 706-714.

(3) Hui, K. P. Y., et al. (2022). "SARS-CoV-2 Omicron variant replication in human bronchus and lung ex vivo." *Nature* 603(7902): 715-720.

It is well known that Omicron variants have altered virus entry pathway, reduced fusogenicity, reduced pathogenicity and altered tissue tropism.

Originally, primarily based on the publication below:

(4) Pastorio, C., Zech, F., Noettger, S., Jung, C., Jacob, T., Sanderson, T., Sparrer, K.M.J., Kirchhoff, F. (2022) Determinants of Spike infectivity, processing, and neutralization in SARS-CoV-2 Omicron subvariants BA.1 and BA.2. *Cell Host Microbe* 30: 1255-1268 e5

We perceived that S371L/F-S373P-S375F mutations have the biggest impact on virus entry and spike processing. Both of above properties have been proposed to link with virus tissue tropism changed and attenuation. Although a recent preprint also shows specifically that G339H, S371F, S373P and S375F are important for authentic virus epithelium cell tropism (Furnon et al., 2024; doi: 10.21203/rs.3.rs-4283987/v1), a few studies now suggest that S371F, S373P and S375F are functional only when co-occured with other Omicron specific mutations (Furnon et al., 2024; doi: 10.21203/rs.3.rs-4283987/v1. Martin et al., 2022; PMID: 35325204). Indeed, S371F, S373P and S375F almost always appear together and together with the other Omicron specific mutations (Martin et al., 2022;

PMID: 35325204). Most recently, other factors have emerged to play potential roles in Omicron tissue tropism change and disease severity, including the use of metalloproteinases and modulation of host interferon responses. In light of these new results, we now conclude that the correlation between SARS-CoV-2 tissue tropism, pathogenicity, immune evasion, and transmissibility is highly complex and remains ill-defined. Further research is likely needed to understand the full exact cause for the reported changes in Omicron phenotypes.

The first two paragraphs of the Discussion section have been revised to reflect the above considerations. (also see our response to Comment Number 1)

3. In the discussion at lines 571-573 the authors state "Our findings further support the hypothesis that convergent antibody responses within the population drive viral antigenic drift leading to emergence of new SARS-CoV-2 variants.", as well as in the abstract at lines 43-45 "These findings support that this class of public antibodies constitutes immune pressure promoting the introduction of S371L/F-S373P-S375F in Omicron variants." and in the title "Antibodies utilizing VL6-57 light chains target a convergent cryptic epitope on SARS-CoV-2 spike protein driving the genesis of Omicron variants". In my opinion, these statements are mainly and exclusively reflected in the experiments conducted in the infected population (Figure 4, see comment number 7 in the results' section). For this reason, I believe that they should reconsider these conclusions.

Response: We thank the reviewer for this thoughtful comment and for pointing out the potential overreaching conclusions drawn from our data.

Please also see our response to Comment Number 7 regarding the result section. Briefly, our findings, along with those previously published, in our opinion, can substantially support our conclusions. In light of the reviewer's feedback, we have carefully re-evaluated our conclusions and agree that a more cautious interpretation of our findings may be warranted. Thus, we have revised the Discussion section to reflect a more measured tone:

"Our findings lend support to the hypothesis that convergent antibody responses may influence viral antigenic drift. However, establishing a direct causal link between immune pressure and the emergence of specific mutations such as those seen in Omicron variants likely requires additional longitudinal and functional studies." (Line 794-798)

Also, we have revised the title to better align with the data:

"Antibodies utilizing VL6-57 light chains target a convergent cryptic epitope on SARS-CoV-2 spike protein and **potentially** drive the genesis of Omicron variants". (Line 1-2)

And modified the abstract as suggested:

"These findings support that this class of public antibodies constitutes **a potential immune pressure** promoting the introduction of S371L/F-S373P-S375F in Omicron variants." (Line 46-48)

We are grateful for the reviewer's guidance in improving the depth and

accuracy of our conclusions.

4. What is the potential application of this discovery in the future development of new generation, more potent COVID-19 vaccines?

Response: We thank the reviewer for this inspiring question. As illustrated in Figure 6b of the manuscript, the RBD surface targeted by VL6-57 antibodies is among the most conserved surfaces on the RBDs of SARS-CoV-2 and related CoVs. In fact, S371 and S375 are highly conserved across most recognized SARS-related CoVs. It has been proposed that this surface could serve as a promising target for pan-sarbecovirus neutralization. However, potentially under immune pressure, this surface is subject to change. This observation suggests that SARS-CoV-2 and related CoVs may possess unrecognized potential for antigenic change in the RBD. Our research results support the idea that other vaccine targeting areas, including the S2 regions, should be investigated or identified.

In addition, the finding that VL6-57 antibodies exhibit a certain degree of cross-reactivity across SARS-related CoVs, and that the convergent epitope is located in a relatively conserved RBD area, will inform the use of this class of antibodies, especially those with broader breadth, for potential emergent infections by SARS-related CoVs. A sentence regarding this point is added in the conclusion:

“We and others also found that some of the VL6-57 mAbs including H18 and many others

(Cao et al., 2023) are able to cross-react with SARS-CoV-1 RBD, confirming that the VL6-57 epitope is relatively conserved among sarbecoviruses and suggesting that cross-reactive VL6-57 mAbs may be used to combat potential emergent sarbecoviruses.” (Line 639-643)

Lastly, our findings highlight the importance of monitoring viral sequences for mutations in key epitopic regions. Future vaccine strategies could benefit from incorporating such surveillance data to pre-emptively adjust vaccine compositions in response to emerging escape variants, ensuring continued vaccine effectiveness. (Line 798-802)

Figure:

1. Figure 2C: The green dashed arrow, indicating a possible direction of glycan chain extension, should be represented in both structures.

Response: We appreciate your suggestion regarding Figure 2C. In the revised 2c, the green dashed arrow, indicating a possible direction of glycan chain extension, have been showed in both structures

2. Figure 2E: Are the comparisons not significant? class 1 mAb S230 results are missing.

Response: We thank the review for pointing out this unambiguity. In the original manuscript, we did not conduct statistical analysis. In the revised manuscript, we have added statistical analysis to ensure clarity on the significance levels.

There may be a confusion regarding the description of mAb S230, which is an antibody targeting SARS-CoV-1 S-protein (we have explained in

detail to the comment 4). Therefore, S230 was not included in the assay here. To avoid any confusion, we have adjusted the figure legend accordingly.

3. Figure S5: 36 structures sharing the characteristic of belonging to class 4 antibodies are shown in Figure S5, however it's hard to analyze them due to the complexity and the huge number of the different structures. The authors should better highlight at least the 11 structures of antibodies encoded by IGLV6-57.

Response: We appreciate the reviewer's suggestion. We have highlighted the 11 structures of antibodies encoded by IGLV6-57 in red boxes for better clarity and focus. We also have revised the labels of the structures in red to highlight these specific structures. These adjustments should help enhancing the readability of the mentioned antibodies within the context of the image.

Reference:

1. Line 960: This is an outdated reference for Dong and associates. The latest manuscript was published in Nature Microbiology, September 2021.

Response: We thank the reviewer for pointing out this issue. We have updated the reference accordingly in the revised manuscript.

Minor points

Introduction:

1. Lines 69-72: In this sentence "recurrent mutations" is repeated.

Response: We thank the reviewer for pointing out this mistake. We have removed the repeated "recurrent mutations".

2. Lines 73-74: It would be better to replace the sentence “..the epitopes of several classes of germline antibodies..” with the new one “..the epitopes recognized by several classes of germline antibodies..”

Response: We thank the reviewer for this suggestion. We have revised the sentence as per your recommendation.

3. Lines 313-318: This sentence is too long and and not easily understood. Please, simplify or break up the sentence.

Response: We thank the reviewer for this suggestion. We have divided the long sentence into smaller sentences for better clarity. The text now reads:

“An LCDR3 sequence analysis reveals that a “QSYDSS” motif is enriched (**Fig. S8d**). A BSA analysis indicates that LCDR3s cover smaller areas by comparison with LCDR1s in VL6-57 mAbs, suggesting that LCDR3s may play an auxiliary role in antigen binding (**Fig. S7c**). By structural analysis, we found that the tyrosine (Y94 in R1-26, **Fig. 2b**) residue within the “QSYDSS” motif is engaging in specific interactions with critical antigen binding residues in HCDR3.” (**Line 402-405**)

REVIEWERS' COMMENTS

Reviewer #1 (Remarks to the Author):

The authors have responded to all concerns.

Reviewer #2 (Remarks to the Author):

The authors adequately answered most of the questions posed. In this way, the manuscript is improved in clarity in th presentation of the results and in the conclusions that have been drawn.

Nevertheless, I have some minor points that should be addressed.

Results:

1. Lines 200-202: the authors state “By contrast, the simultaneous binding of CR3022 and ACE2 is likely possible due to a much weaker clash.” How did the authors define “much weaker clash”? This “conclusion” seems to have been assumed. I would have expected either the presence or the absence of a clash.
2. Lines 185-189 and 211-215: These two sentences are too long and not easily readable. Please, simplify or break up the two sentences.

REVIEWERS' COMMENTS

Reviewer #1 (Remarks to the Author):

The authors have responded to all concerns.

Reviewer #2 (Remarks to the Author):

The authors adequately answered most of the questions posed. In this way, the manuscript is improved in clarity in the presentation of the results and in the conclusions that have been drawn.

Nevertheless, I have some minor points that should be addressed.

Results:

1. Lines 200-202: the authors state “By contrast, the simultaneous binding of CR3022 and ACE2 is likely possible due to a much weaker clash.” How did the authors define “much weaker clash”? This “conclusion” seems to have been assumed. I would have expected either the presence or the absence of a clash.

2. Lines 185-189 and 211-215: These two sentences are too long and not easily readable. Please, simplify or break up the two sentences.

Reviewer #1 (Remarks to the Author):

The authors have responded to all concerns.

Response: We thank the reviewer's constructive comments which helped us to improve the manuscript.

Reviewer #2 (Remarks to the Author):

The authors adequately answered most of the questions posed. In this way, the manuscript is improved in clarity in the presentation of the results and in the conclusions that have been drawn.

Response: We thank the reviewer's constructive comments which helped us to improve the manuscript.

Nevertheless, I have some minor points that should be addressed.

Results:

1. Lines 200-202: the authors state "By contrast, the simultaneous binding of CR3022 and ACE2 is likely possible due to a much weaker clash." How did the authors define "much weaker clash"? This "conclusion" seems to have been assumed. I would have expected either the presence or the absence of a clash.
Response: We thank the reviewer for this insightful comment. We have revised the conclusion as: "By contrast, CR3022 and ACE2 are able to bind the RBD simultaneously without clashing (Fig. S4b)."

2. Lines 185-189 and 211-215: These two sentences are too long and not easily readable. Please, simplify or break up the two sentences.

We thank the reviewer for this helpful suggestion. We have broken up the two sentences.

The revised sentences will be read as: CR3022 is a well-studied class 4 mAb which cross-reacts to RBDs of SARS-CoV-1 and SARS-CoV-2 (Yuan et al., 2020b). We found that although the epitope of R1-26 appears to largely overlap with that of CR3022, R1-26 features a different approach angle towards the RBD compared to CR3022 (Fig. S3a, g and h). (Lines 178-181)

and

This analysis shows that R1-26 possesses an activity to trigger fusogenic conformational change similar to class 1 mAbs including B38 (He et al., 2022) (Fig. 2c) and S230 (Walls et al., 2019). Interestingly, the class 4 antibody CR3022 does not have this triggering activity (Fig. 2c). (Lines 202-205)